



# Variability Scaling and Consistency of Airborne and Satellite Altimetry Measurements of Arctic Sea Ice

Shiming Xu[1,2], Lu Zhou[1], and Bin Wang[1,3]

[1]Ministry of Education Key Laboratory for Earth System Modeling, Department of Earth System Science, Tsinghua University, Beijing, China
[2]University Corporation for Polar Research, Beijing, China
[3]State Key Laboratory of Numerical Modeling for Atmospheric Sciences and Geophysical Fluid Dynamics (LASG), Institute of Atmospheric Physics, Chinese Academy of Sciences, Beijing, China

*Correspondence to:* Shiming Xu (xusm@tsinghua.edu.cn)

**Abstract.** Satellite and airborne remote sensing provide complementary capabilities for the observation of the sea ice cover. However, due to the differences in footprint sizes and noise levels of the measurement techniques, as well as sea ice's variability across scales, it is challenging to carry out inter-comparison or consistency study of these observations. In this study we focus on the remote sensing of sea ice thickness parameters, and carry out: (1) the analysis of variability and its statistical scaling

for typical parameters, and (2) the consistency study between airborne and satellite measurements. By using collocating data between Operation IceBridge and CryoSat-2 in the Arctic, we show that there exists consistency between the variability of radar freeboard estimations, although CryoSat-2 has higher noise levels. Specifically, we notice that the noise levels vary among different CryoSat-2 products, and for ESA CryoSat-2 freeboard product the noise levels are at about 14 and 20 $cm$ for first-year and multiyear ice, respectively. On the other hand, for Operation IceBridge and ICESat, it is shown that the variability

of snow (or total) freeboard is quantitatively comparable, despite over 5 years' the time difference between the two datasets. Furthermore, by using Operation IceBridge data, we also find wide-spread negative covariance between ice freeboard and snow depth, which only manifest at small spatial scales (40 $m$ for first-year ice and about 80 to 120 $m$ for MYI). This statistical relationship highlights that the snow cover reduces the overall topography of the ice cover. Besides, there is prevalent positive covariability between snow depth and snow freeboard across a wide range of spatial scales. The variability and consistency

analysis calls for more process-oriented observations and modeling activities to elucidate key processes governing snow-ice interaction and sea ice variability on various spatial scales. The statistical results can also be utilized in improving both radar and laser altimetry, as well as the validation of sea ice and snow prognostic models.



## 1 Introduction

Sea ice and its snow cover is an integral component of the earth's climate system. Basin-scale Arctic sea ice concentration observations have been available since 1978 with passive microwave satellite remote sensing (Cavalieri et al., 1999). During this period, the Arctic sea ice cover has undergone drastic changes, with record-lows of September extent minimums as a most prominent feature. Accompanying the all-season shrinkage of the Arctic sea ice cover are the overall thinning of the sea ice (Stroeve et al., 2014), as well as the transition to younger ice age (Lindell and Long, 2016). Besides, the thermodynamics of sea ice and the polar air-sea interaction is greatly modulated by the snow over the sea ice (Webster et al., 2018). Due to snow's low thermal conductivity and high albedo, it can effectively insulate air-sea heat exchange and play important roles in the positive albedo feedback. With climate warming, there are also growing evidences of changes in snow properties (Webster et al., 2014). However, there still exist large gaps in understanding snow processes, especially its interaction with sea ice, mainly due to limited observations and deficiencies of sea ice and climate models. Sea ice, together with its snow cover, is a focus for the international research community, from both observational and modeling perspective.

Among various sea ice parameters, the thickness parameters, including sea ice thickness ($h_i$) and snow depth ($h_s$), are essential to sea ice related climate research and key applications. Ice thickness is a direct indicator of the history of both thermodynamic and dynamic interaction between polar atmosphere and ocean. Due to its longer persistence, ice thickness and volume can be potentially utilized to improve forecasts on seasonal or longer scales (Chen et al., 2017; Blockley and Peterson, 2018). However, despite their importance, thickness parameters are more challenging for observations in both in-situ campaigns and remote sensing. Satellite altimetry is the major approach for the estimation of sea ice thickness at basin-scale. By sending active signals from the satellite to the earth's surface and measuring the latency of backscattered signals, satellite altimetry determines the range between the satellite and the scattering plane of the signal on the earth. This range is converted into the height information, and by differentiation of echoes on ice floe from those on water (i.e., leads), and retracking of lead and floe height. Reconstruction of the local water level is carried out, based on water levels in leads and environment conditions (large-scale dynamical height, tidal effects, atmospheric loading, etc). The correction for local sea-surface height (SSH) is then added to the floe's height to retrieve the freeboard, which is the difference between the range of floes and that of the reconstructed local water level.

Fig. 1.a shows the typical parameters of thickness retrieval of sea ice, including satellite altimetry. There are mainly two types of satellite altimetry: laser altimetry and radar altimetry. For laser altimetry (Kwok and Cunningham, 2008), the main backscattering plane mainly resides close to the surface of the snow cover, and the main target is the retrieval of the snow freeboard ($F_s$). For Ku-band radar altimetry such as CryoSat-2 (Wingham et al., 2006), the backscattering mainly occurs within the snow cover, and it is usually assumed that radar signals effectively penetrate the snow cover. According to Armitage and Ridout (2015), there is overall 82% penetration into snow over multi-year ice (MYI) and 97% over first-year ice (FYI). Due to slower penetration speed ($C_s$) of radar signal in the snow than in the air, the "raw" range includes a bias which should be accounted for by a correction term determined by $C_s$ and $h_s$. This raw elevation before correction is denoted radar freeboard ($F_r$), while the corrected freeboard ice freeboard ($F_i$). Under the assumption of climatological snow density of 320 $kg/m^3$,



the correction term is approximately 1/4 of $h_s$ . Fig. 1.a shows the general case of limited penetration (effective penetration depth of $h_s{}^*$), and the correction term should be in turn associated with $h_s{}^*$ (instead of $h_s$ ) for the non-biased elevation of the main reflection plane.

The freeboards are in turn converted into ice thickness estimations, under the assumption of hydrostatic equilibrium and
buoyancy relationships (Eqs. 1 and 2). This conversion depends on accurate estimations of the following parameters: snow depth, snow density ($\rho_s$), ice density ($\rho_i$) and water density ($\rho_w$). In existing CryoSat-2 based products, climatological snow depth and density based on Warren et al. (1999) are usually adopted for this conversion, as well as for the correction term of slow radar propagation. For existing ICESat products, snow depth fields are reconstructed based on accumulation of reanalysis-based precipitation and numerical sea ice drifts (Kwok and Cunningham, 2008). For both types of altimetry, snow properties
remains a major source of uncertainty in the retrieval of $h_i$ , while other factors including ice density also play important roles in determining the overall uncertainty (Zygmuntowska et al., 2014; Tilling et al., 2015).

$$h_i = \left(\frac{\rho_w}{\rho_w - \rho_i}\right) \cdot F_i + \left(\frac{\rho_s}{\rho_w - \rho_i}\right) \cdot h_s \tag{1}$$

$$h_i = \left(\frac{\rho_w}{\rho_w - \rho_i}\right) \cdot F_s - \left(\frac{\rho_w - \rho_s}{\rho_w - \rho_i}\right) \cdot h_s \tag{2}$$

Airborne surveys provide high-resolution scanning of the sea ice cover, which usually feature more payload types and have
complementary observational capabilities with satellites. They also provide invaluable calibration and validation support for satellite retrieval. NASA's Operation IceBridge (OIB) and ESA's CryoSat Validation Experiment (CryoVEx) are representative airborne campaigns which provide both scientific evidences of sea ice parameters and practical support to satellite altimetry. For OIB, total freeboard and snow depth are retrieved, and sea ice thickness can be derived with altimetric relationships (Eqs. 2, see also Fig. 1.a). Commonly available on CryoVEx campaigns is the electromagnetic induction sensor (EM) which is towed
under the fixed-wing platform, and the total thickness of snow and ice ($h_i + h_s$) is retrieved.

Both OIB and CryoVEx have limited coverage of the sea ice cover, and the measurements are concentrated along the flight tracks. Nominally, the sea ice thickness and snow depth from OIB products has approximately 40-$m$ resolution (see Sec. 2.1.1 for details). For CryoVEx, the footprint and cross-track coverage of airborne EM is about 50 to 70 $m$. Various existing works have compared freeboard retrieval of CS-2 against OIB, and there is usually very low statistical correlation even for collocating
tracks (Kurtz et al., 2014; Xia and Xie, 2018; Yi et al., 2018). On the other hand, by adopting the same the geophysical corrections of CS-2, Yi et al. (2018) effectively aligns the retrieved freeboard across CS-2 products and greatly reduces the systematic differences. The limited representation of OIB as compared with CS-2 due to the relatively small coverage of OIB is mainly attributed as the cause of the low correlation.

In this study, we investigate the variability and its scaling among airborne and satellite remote sensing of thickness pa-
rameters. The parameters subjected to analyses include snow depth, radar freeboard and snow freeboard. The variability (in terms of variance and standard deviation) of a certain parameter is essentially governed by its inherent, physical variability. However, the estimation of variability through sampling is subjected to the footprint of observations and measurement errors (or noise levels) which are specific to each sensor/campaign. Therefore, we account for data product uncertainties during the





analysis of the variability and its scaling. In order to avoid the extra uncertainties in ice thickness retrievals (introduced during the altimetric relationships), we analyze freeboard instead of ice thickness. The collocating data between CS-2 and OIB since 2011 during high winters of the Arctic are used for the analysis. Furthermore, data from collocating tracks between CS-2 and airborne campaigns of OIB and CryoVEx are utilized. For laser altimetry, we adopt ICESat and study the statistical behavior of

variability and compare with OIB (due to no available collocating data). Section 2 includes details of the dataset of satellite and airborne campaigns, and the specific treatments and methods for analysis. Section 3 covers all the results, including: analysis with collocating measurements between CS-2 and OIB, analysis of statistics of scaling for OIB and ICESat, and covariability analysis based on OIB dataset. In Section 4 we summarize the article and discuss related topics including the effects of variability and covariability on sea ice altimetry and snow-ice interaction.

## 2    Data and Methods

In this study, we focus on thickness related parameters measured by airborne and satellite campaigns for the Arctic sea ice. In specific, the following datasets are used: (1) OIB datasets of 40-$m$ scale snow depth, snow freeboard, derived ice freeboard and radar freeboard; (2) CS-2 (ESA) per-sample radar freeboard that collocates with OIB; (3) ICESat per-sample total freeboard during Feb., Mar., and Apr.; and (4) AEM measured total thickness of ice and snow from CryoVEx (collocating tracks with

CS-2). Sec. 2.1 gives detailed introduction to these datasets, and Sec. 2.2 contains the necessary treatments for analyses and inter-comparison.

   Before the analysis, we also formally define the physical parameter and its measurement error as follows. We denote the measurement of any parameter $a$ as $a|_{obs}$ which contains the linear combination of the inherent, physical status ($a|_{phy}$) and uncertainty terms, including the systematic bias ($e$) and the random error ($\epsilon$). Adopting both $e$ and $\epsilon$ allows us to differentiate

the behavior of these two types of uncertainty during scaling analysis.

$$a|_{obs} = a|_{phy} + e + \epsilon \tag{3}$$

   Biases ($e$) arise from both measurements and treatments to the measurements. As an example, sea-surface height correction in altimetry is computed from local water level estimations based on sea ice lead detection. The retracking error in sea ice leads causes uncertainty in the freeboard estimation of the sea floes. Therefore, the freeboard uncertainty that is associated with SSH

correction are usually persistent across adjacent altimetric samples, and at local scales it is treated as a bias. Biases affect the estimation of mean value of the parameter, but not its second-order statistics (i.e., variance and standard deviation). On the other hand, random errors ($\epsilon$) which usually arise from measurements and limited by the sensors' precision, are independent between samples and usually follow normal distributions. During the analysis of variability, we only consider random errors, and ignore the contribution from biases. With scaling, random errors usually diminishes fast through averaging. Based on these

formulations, we carry out the sample-based analyses of variability and scaling (details in Sec. 2.3).



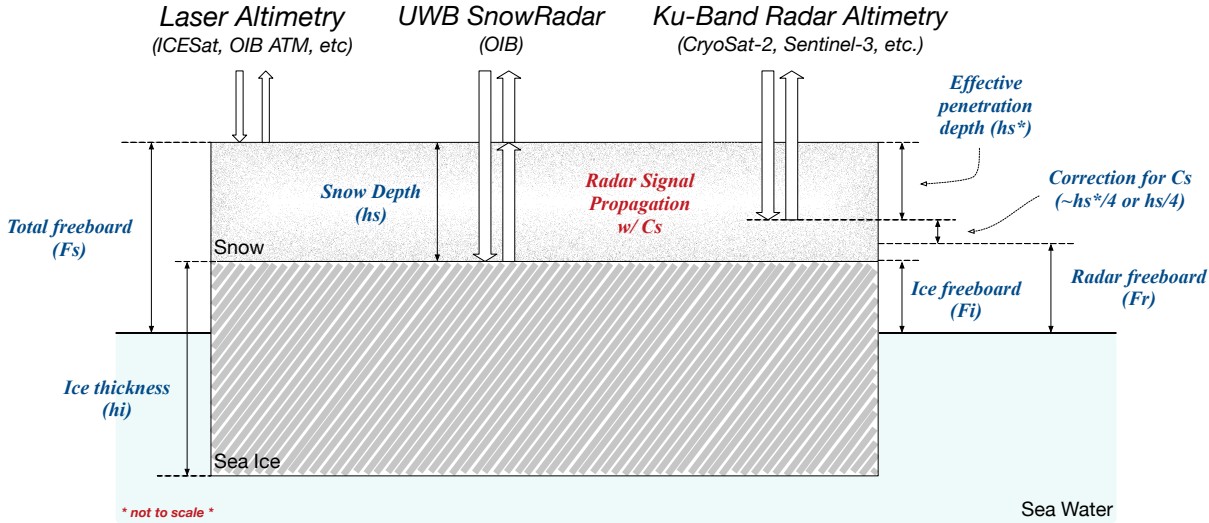

(a) Remote sensing of thickness parameters of sea ice

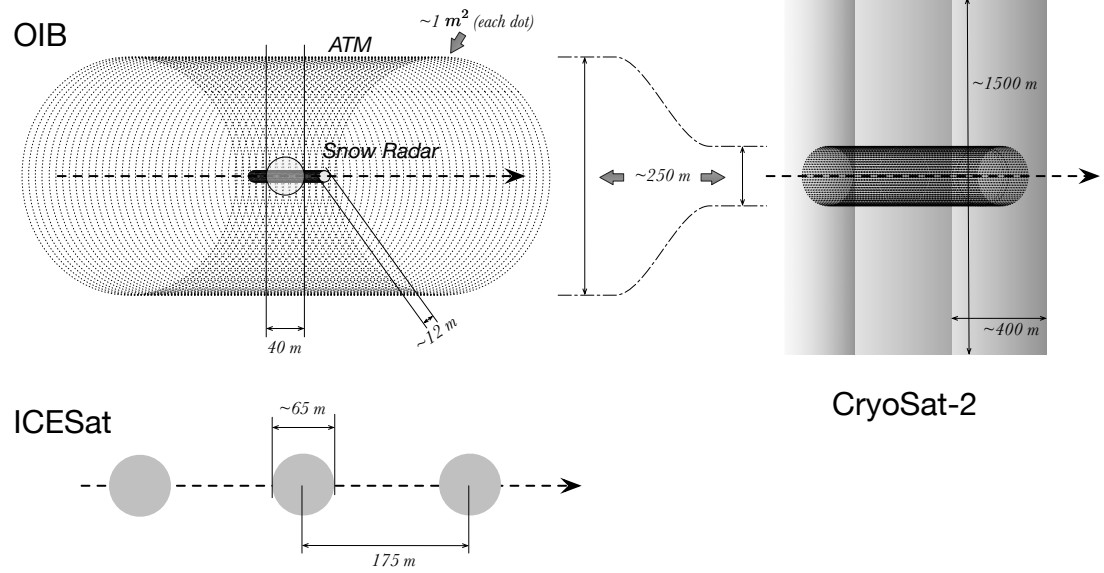

(b) Typical footprint and resolution of OIB, CryoSat-2 and ICESat

**Figure 1.** Sea ice remote sensing by CryoSat-2, ICESat and Operation IceBridge.





## 2.1 Datasets

### 2.1.1 Operation IceBridge

Since 2009, NASA's Operation IceBridge (OIB) has been carrying out surveys with fixed-wing airborne remote sensing in the western Arctic during high winter months (mainly around March and April). In each campaign, the sea ice cover along the

flight path is scanned with various onboard sensors, yielding high-resolution measurements of sea ice parameters, including freeboard, snow depth, visual images, etc. The major device onboard is the Airborne Topographic Mapper (ATM) which utilizes conic scans with laser beams (Krabill, 2009). The coverage of the ATM on the ground is spiral, progressive scans centered near the flight path, with: (1) each laser footprint at about $1\ m^2$, and (2) a swath of about $250\ m$ for wide-swath setting at the nominal flight heigh of about $460\ m$ (as in Fig. 1.b). Under wide swath scanning, the nominal distance between each point on the nadir

of the path is about 2 to $3\ m$. For certain campaigns, narrow swath scanning is available, which increases the footprint density at nadir of the flight. Through visual inspection of imagery from onboard Digital Mapping System (DMS) and differentiation between reflections from leads (water or very thin ice) and floes, the elevation of the sea ice floes (i.e., total freeboard, $F_s$) is retrieved (Kurtz et al., 2013).

Another sensor onboard for thickness parameter retrieval is the ultra-wide band snow radar (SnowRadar) from University

of Kansas (Leuschen, 2014). SnowRadar periodically sends $2\ GHz$ to (about) $7\ GHz$ wide-band microwave signals to the sea ice cover, and records backscattered waveforms. By retracking the major scattering planes in the waveforms, the travel latencies between air-snow interface and snow-ice interface are tracked down, and the snow depths are retrieved under certain assumptions of snow density and radar travel speed in snow (Kurtz et al., 2013). The nominal footprint size of SnowRadar (with flight altitude at about $460\ m$) is $11\ m$ across track and $14.5\ m$ along-track on snow-covered sea ice (Kurtz et al., 2013),

with a minimum detectable snow depth of $5\ cm$.

OIB campaigns date back to 2009, and in this study, we carry out analysis based on two OIB datasets that contain campaigns between 2011 and 2017. The first is the IceBridge L4 Sea Ice Freeboard, SnowDepth, and Thickness (IDCSI4) product for OIB campaigns between 2011 and 2013 (Kurtz et al., 2015). Since this product does not contain campaigns after 2013, we also use the IceBridge Sea Ice Freeboard, Snow Depth, and Thickness Quick Look (Kurtz et al., 2012, updated each year) for campaigns

between 2014 and 2017. In these products, measurements of $h_s$ are averaged within each 40-$m$ segment (about 50 SnowRadar samples), in order to reduce the noise level of individual SnowRadar footprint. In order to combine the measurements by ATM and $h_s$ to generate ice freeboard ($F_i$) and ice thickness, all the ATM height measurements within $20\ m$ of the center of the SnowRadar measurements are averaged to produce $F_s$ (see the hollow circle in Fig. 1.b). Ice freeboard then is derived as: $F_i = F_s - h_s$. In turn, ice thickness can be computed using typical buoyancy relationship widely adopted in altimetry.

The measurement accuracy of independent ATM scans is about $3\ cm$ (Martin et al., 2012), and usually over 200 ATM samples are averaged to produce $F_s$ at 40-$m$ scale. With averaging, the random error of $F_s$ due to ATM measurement errors is very small. The uncertainty of $F_s$ is further determined by factors including available SSH observations within the local regions, which are variable along the track and in the range from $1\ cm$ to $30\ cm$ (Kurtz et al., 2015). Since SSH height information are shared among freeboard data, we treat this uncertainty as bias and ignore it in the scaling analysis. The uncertainty of snow





radar is inherently limited by its range resolution of about 5 $cm$ after windowing. The overall uncertainty of $h_s$ of OIB product is estimated to be 5.7 $cm$ through validation with in-situ data (Kurtz et al., 2013). In this study we adopt 5 $cm$ as the random error associated with $h_s$ .

### 2.1.2 CryoSat-2 (CS-2)

The European Space Agency's (ESA) satellite campaign CryoSat-2 has been monitoring the Arctic sea ice cover since autumn of 2010. Onboard CryoSat-2 is the delay-doppler Ku-band radar altimeter SIRAL (Parrinello et al., 2018). By delay-doppler treatment of pulse-limited radar signals and range tracking, CryoSat-2 achieves the nominal resolution of about 400 $m$ by 1500 $m$ (see Fig. 1.b), which greatly enhances that of conventional pulse-limited radar altimeters (Resti et al., 1999). Lead detection, lead/floe retracking, SSH correction is then carried out to convert L1 stacked waveforms to L2 $F_r$ . Then $F_r$ is converted into

$F_i$ with radar propagation speed and snow depth estimations (Fig. 1.a and Eqs. 4). For CS-2, it is usually assumed that radar signal fully penetrates the snow cover and the major scattering horizon is at the snow-ice interface. However, there are growing evidence that the effective backscattering plane may be shifted upward, mainly due to scattering within the snow cover (Ricker et al., 2015; King et al., 2018; Nandan et al., 2017). In the general case of limited penetration (penetrated depth $h_s{}^*$ smaller than the true $h_s$ ), the correction term associated with radar propagation speed should be changed accordingly (Fig. 1.a) .

$$F_r = F_i - \left(1 - \frac{c_s}{c}\right) \cdot h_s \approx F_i - 0.25 \cdot h_s \qquad (4)$$

    It is worth to note that there exist large differences of the L2 production protocols to produce ice freeboard, including ESA, CPOM, AWI, among others. The differences mainly fall into 4 categories: (1) lead detection, (2) lead and floe retracking, (3) snow depth and its correction term, and (4) SSH correction. For example, all these three products adopt threshold based floe retracker, but differ in terms of the specific threshold value (50% for ESA and AWI, and 70% for CPOM). Another example

is that the correction for slow radar propagation in snow cover for AWI (Ricker et al., 2014) and CPOM (Tilling et al., 2017) are based on different climatological snow density and snow depth settings (Warren et al., 1999), while this correction is not present in ESA's product (baseline C). Despite these differences, as shown in Yi et al. (2018), under same protocols for geophysical corrections, there is general consistency (within 5 $cm$) for the mean freeboard among these datasets. Beside, we have also found that the correlation between the along-track freeboard measurements among these products are also very high

(not shown).

    In this study, we mainly use ESA's CS-2 L2 radar freeboard product (baseline-C) for analysis. We also adopt AWI's L2 ice freeboard product for comparative studies, but carry out de-correction of the radar propagation speed to deduce the radar freeboard, following AWI's protocol (Ricker et al., 2014). Since altimetric scans (such as CS-2) only cover the nadir of the satellite's ground path, it usually requires monthly measurements to achieve basin-scale coverage. Therefore, we use along-

track freeboard data, and compile them when needed (see below for details). The freeboard uncertainty associated with speckle noise is estimated at about 10 $cm$ (Wingham et al., 2006). Besides, in AWI's CS-2 protocols, the uncertainty associated with SSH correction is in the range of 5 to 50 $cm$, while the bias caused by the fixed retracking threshold and limited penetration is



estimated to be 6 and 12 $cm$ for FYI and MYI, respectively (Ricker et al., 2014). Since the uncertainty associated with SSH is dependent on lead detection and specific treatments of along-track interpolation, its contribution to systematic error ($e$) and random error ($\epsilon$) is also variable. For example, in Tilling et al. (2017), 100 $km$ is chosen as the range of valid lead observations for determining the local SSH. Therefore, the uncertainty associated with SSH at adjacent $F_r$ samples along each track is highly

correlated, but it will be much more dependent on longer spatial ranges (e.g., over 100 $km$). This scale is usually much larger than the scaling analysis in this study (usually within 2 $km$, see Sec. 3). Therefore, in this study, SSH related uncertainties in freeboard measurements are treated as systematic error and ignored in the scaling analysis.

### 2.1.3 CryoSat Validation Experiment (CryoVEx)

Another airborne campaign dataset we compare against satellite data is CryoVEx. Onboard sensors of CryoVEx include the

airborne electromagnetic induction sensor (AEM) and laser scanner, and total thickness of snow and ice ($h_i + h_s$) is retrieved. The effective resolution (footprint) by AEM is about 50 to 70 $m$ (Haas et al., 2010, 2009), with an accuracy of 0.1 $m$ on level ice. In order to produce correspondence with CS-2 measurements, the flight lines of CryoVEx campaigns collocated with ground tracks of CS-2. Since there are relatively smaller overall coverage from CryoVEx campaigns, we use the available CryoVEx data together with OIB campaigns (only collocating tracks with CS-2) for certain analysis. Specifically, we use

dataset provided by ESA, which contains CryoVEx campaigns in 2011, 2012 and 2014.

### 2.1.4 ICESat

Between 2003 and 2009, NASA's Ice, Cloud, and land Elevation Satellite (ICESat) carried out remote sensing of earth surface's elevation with its onboard laser altimeter. The ground track of ICESat consists of illuminated regions of 65 $m$ in diameter, with consecutive ground footprints along flight track about 175 $m$ apart (Kwok and Cunningham, 2008). With lead detection and

SSH estimations, the snow freeboard ($F_s$) is retrieved. In turn, under certain knowledge of snow depth and snow/ice density (Kwok and Cunningham, 2008), sea ice thickness is attained (Eqs. 2). Wintertime campaigns over the Arctic sea ice yields basin-scale ice thickness fields on the bi-monthly basis.

Since there is no temporally collocating data between ICESat and OIB, we carry out the analysis of statistical scaling and its consistency with respect to ice types (MYI or FYI). In specific, we use ICESat along-track $F_s$ product (Yi and Zwally, 2009)

for campaigns during high winters (February and March, or March and April) for analysis.

The precision of GLAS sensor is estimated to be several centimeters, and in this study we adopt 5 $cm$ as the noise level for each ICESat footprint. As a reference, according to Kwok and Cunningham (2008), the 25-$km$ segment mean $F_s$ has the uncertainty of about 5 $cm$, which is inherently limited by tie-points during the production of $F_s$.

### 2.2 Data summary and treatments

Fig. 2 shows all the geolocations of OIB and CryoVEx data as used in this study. Although OIB flight lines achieves large-scale coverage, the actual area of Arctic sea ice cover as scanned by OIB is very small. In Fig. 2 we also show 3 local regions with

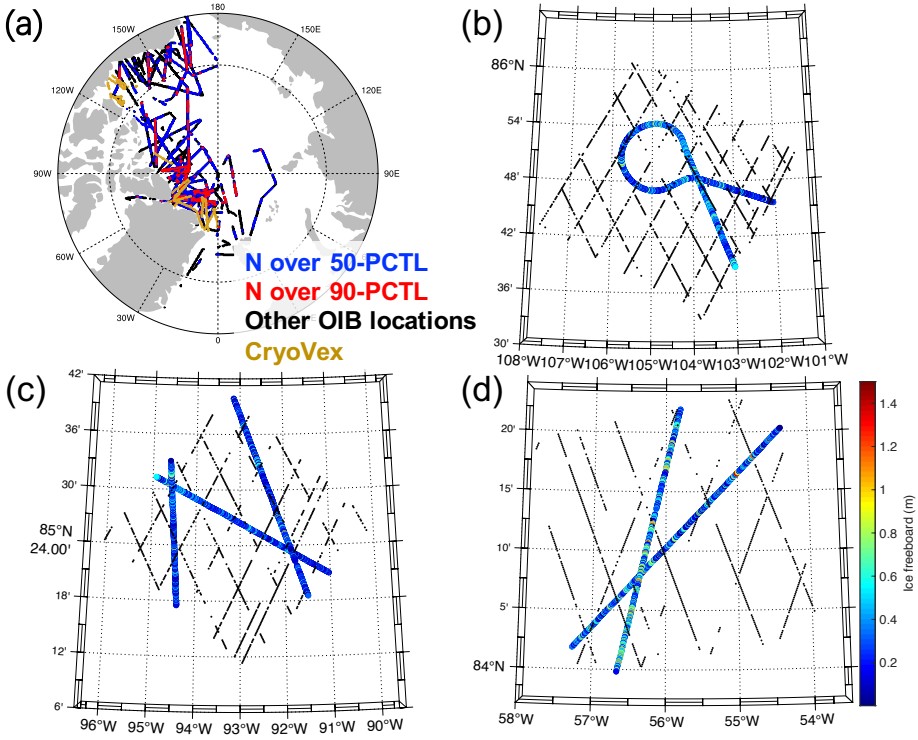

**Figure 2.** Locations of OIB and CryoVEx measurements from 2011 to 2017 (a), with sample regions of good OIB coverage (b to d). OIB measured ice freeboard values are shown in color, with effective ESA CS-2 samples within the same month of OIB campaign shown by black dots.

good OIB coverage. Each region corresponds to $3 \times 3$ EASE grid cells and has area of $37.5 \times 37.5 \ km^2$. For these regions, the OIB flights contain recursive fly-overs or cross-over points. The valid ESA CS-2 sample points are shown for those within: (1) the same month of the corresponding OIB campaigns, and (2) the same EASE grid cells.

For each local region within the Arctic, we record the effective OIB sample count (40-$m$ scale), denoted $N$, as an indicator for OIB coverage at local regions. The value of $N$ follows a long-tail distribution (not shown) for all the local regions, and in Fig. 2.a we show with different colors the regions with relatively better OIB coverage ($N$ of the local region over 50-th and 90-th percentile for all local regions). $N$'s for the 3 regions in Fig. 2 are 1'856, 2'389 and 1'907, which are all over 99-th percentile for $N$. Due to the sparse OIB coverage even at local regions, in order to improve the representativeness of OIB and avoid limited OIB sample count, we limit the analysis of OIB to the regions with relatively good OIB coverage ($N$ over 50-th percentile).





### 2.2.1 Local regions as a basis for analysis

The statistical analyses in this study are mainly carried out on local regions ($37.5 \times 37.5 \ km^2$). For any type of measurement (OIB, CryoVEx, CS-2 or ICESat), we use the samples (usually organized in tracks) for each local region to carry out scaling analysis. Basin-scale analysis is further carried out by using the statistics at local regions as samples. The purpose of using local regions is to study the behavior of different measurements within a small region, which usually have relatively homogeneous sea ice cover. Besides, since airborne measurements are relatively scarce, adopting larger scales (e.g., over 100 $km$) will further deteriorate the representativeness of the underlying sea ice cover. This spatial scale (37.5 by 37.5 $km^2$) is also on par with the typical resolution by satellite passive microwave remote sensing, as well as the scale adopted by many gridded altimetry products.

When comparing OIB with CS-2 or ICESat, in order to increase the correspondence of satellite data to (daily) airborne campaigns, we adopt all samples from the same month for CS-2 (or bi-monthly data from ICESat) for each local region for the scaling analysis. On the other hand, for collocating tracks between airborne campaigns (OIB or CryoVEx) and CS-2, we only use CS-2 measurements on the same tracks for analysis.

### 2.2.2 Treatments to $F_r$

In order to reduce the uncertainty, we choose ice freeboard instead of ice thickness for comparing satellite and airborne data. Furthermore, for the comparison between OIB and CS-2, we use $F_r$ instead of $F_i$. Since $F_i$ in CS-2 products contain potentially different and incoherent snow corrections, we convert CS-2 $F_i$ back into $F_r$ when snow propagation has been applied in the product. In order to align OIB data with this treatment, $F_r$ is simulated for each OIB 40-$m$ sample, based on $F_i$ and $h_s$ provided by OIB (Eqs. 4). This equivalent radar freeboard by OIB takes into account the effect of slow propagation of radar in the snow cover, and we assume total penetration of radar signal in the (OIB-measured) snow depth. Besides, we also analyze $F_i$ from OIB when it is needed. For the study with OIB and ICESat, we simply use $F_s$ samples from both datasets for analysis and comparison.

### 2.3 Methods for scaling analysis

For a local region of $37.5 \times 37.5 \ km^2$ (9 EASE grid cells), we carry out the following analysis. For OIB, we locate segments of OIB tracks that are: (1) from the campaigns within the same year and month, and (2) within the local region. For CS-2, we locate segments of CS-2 tracks that are: (1) from the same month as the campaign, and (2) within the local region. For ICESat, the treatment is similar with CS-2, utilizing bi-monthly ICESat tracks for each local region. As is shown in Fig. 1 and Sec. 2.1, there is large difference between footprint and coverage among the remote sensing techniques, as well as uncertainties in measurements. For each local region, we consider the measurements from airborne campaigns and those from satellite collocating, and carry out the scaling analysis.

At each local region, for a certain parameter (e.g., $F_r$ from OIB and CS-2, or $h_s$ from OIB), the analysis mainly involves analyzing the parameter's variability and the change of its variability at coarser spatial scales. The variability at larger scales





beyond the original resolution is estimated by: (1) computing the locally-averaged values of the parameter based on samples, and (2) estimating the sample variance (VAR) or standard deviation (STDEV) from the (locally averaged) values. The sample count for local averaging is denoted by $M$. When $M = 1$, the original resolution is adopted (without averaging). In order to ensure enough sample count for estimating variability when $M$ is large, we limit the analysis involving monthly altimetry

tracks to regions with OIB sample count $N$ larger than 709 (over 50-th percentile, see Fig. 2.a for details). This allows over 30 samples even if using $M = 20$ (800 $m$) for local averaging with OIB data.

If the parameter subjected to scaling analysis is independent and follows the same distribution within the region of analysis, the sample variance should decrease at the speed of $1/M$ (or $1/\sqrt{M}$ for STDEV). However, other factors may modulate the scaling of variability, including: (1) spatial correlation in adjacency (autocorrelation), (2) the inhomogeneity of the sea ice cover

within the region of study. Nonnegative auto-correlation and inhomogeneity would usually cause slower decrease of variability under scaling. The faster the decrease speed is (approaching $1/\sqrt{M}$ for STDEV), the more homogeneous the parameter is within the region of study. Besides the local averaging based analysis, we also adopt a "randomized sampling" strategy: $M$ randomly chosen samples of the local region are averaged (instead of adjacent samples) to compute the statistics of STDEV and VAR. Since with random samples, the effects of both autocorrelation and inhomogeneity are very limited, the behavior of

STDEV (or VAR) with scaling is expected to follow the assumption of independent variables ($1/\sqrt{M}$ decrease in STDEV).

On the other hand, random errors ($\epsilon$) in the measurements of the physical parameters would also affect variability analysis. Specifically, they are assumed to be: (1) additive to the true physical value, (2) following Gaussian distribution, (3) independent from measurement to measurement, and (4) independent from the true value of the physical parameter. Under these assumptions, the sample-estimated variability includes an additive term arisen the random error, and this term decreases with $M$ ($1/M$

for VAR). Therefore, if slow decrease in VAR (or STDEV) is witnessed during scaling, it can be induced that the inherent properties of the physical parameter, rather than random error due to measurements, is the major cause.

## 3  Results and analysis

### 3.1  Analysis of sample regions

We start with analysis for the 3 regions with good OIB coverage (Fig. 2). By using OIB and CS-2 samples in these regions,

we compare the scaling of $F_r$ and $h_s$ as measured by OIB, and CS-2 measured $F_r$ . With randomized sampling, the sample standard deviation (STDEV) decreases with the square root of sample count $M$ (or $\sqrt{M}$) as used for averaging for both OIB and CS-2 (Fig. 3.a). However, CS-2 shows overall much larger variability than OIB: (1) on the original resolution for both OIB and CS-2, STDEV of $F_r$ is already larger in CS-2 than OIB; (2) on the scale of 400 $m$ (CS-2 footprint size in the along-track direction), STDEV of $F_r$ of OIB is lower than 1/3 of that of CS-2. It is worth to note that at 400 $m$ scale (with local averaging of

10 OIB samples of 40 $m$), the effective OIB footprint is still much smaller than (about 1% of) CS-2. With equivalent footprint size, the STDEV as measured by OIB is expected to be much smaller than that of CS-2.

Based on local averaging, STDEV of $F_r$ is also smaller in OIB than in CS-2 (Fig. 3.b). However, for both OIB and CS-2, the decrease speed of STDEV with respect to $M$ is much lower, especially for OIB. For OIB, the slopes of STDEV decrease is





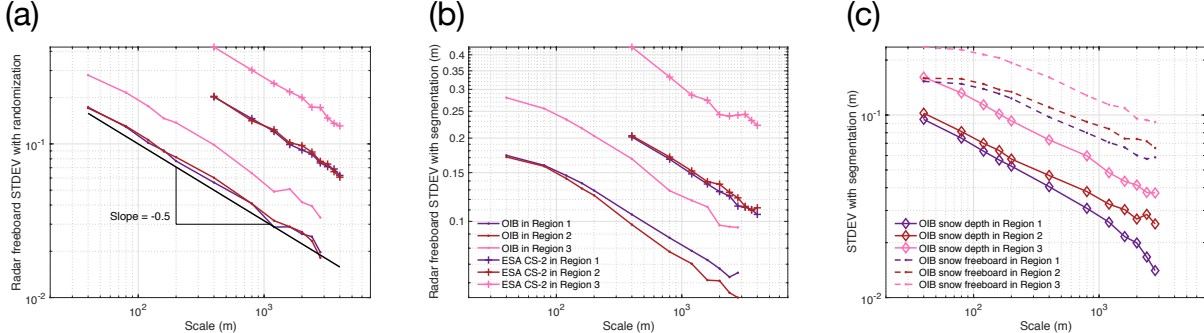

**Figure 3.** Statistical scaling of sea ice measurements by OIB and CS-2 for sample regions in Fig. 2. Scalings of $F_r$ under randomized sampling (or local averaging) by OIB (dots) and CS-2 (+) are shown in a (or b). Scalings of OIB measured $h_s$ (diamond) and $F_s$ (dashed) are shown in c.

about -0.24, -0.27 and -0.27 for these 3 regions, respectively. For CS-2, the decreasing speeds are slightly higher, with slopes of -0.28, -0.26 and -0.33. This indicates that at for both OIB and CS-2 measured $F_r$ , there exists large spatial variability at both the local scale (within several hundreds of meters) and larger scales.

The relatively higher variability of CS-2 $F_r$ is suspected to be due to large noise level of CS-2 at the per-sample scale. The
STDEV of uncertainty due to speckle noise, SSH correction and various other sources are estimated to be larger than 10 $cm$ (Ricker et al., 2014). Therefore, gridding is usually carried out to generate monthly CS-2 sea ice thickness products to improve both spatial coverage and reduce noise (Laxon et al., 2013). By aligning OIB with the along-track footprint size of CS-2 at 400 $m$, we show that the difference of $F_r$ variability (STDEV) between CS-2 and OIB is in range of 20 to 40 $cm$. This comparison of uncertainty is then qualitatively consistent with previous studies. Systematic analysis with all OIB data is further carried out
in Sec. 3.2.

For these 3 regions, we also compare the scaling of $h_s$ and $F_s$ in Fig. 3.c with local averaging. Compared with $F_r$ by OIB (Fig. 3.b), $F_s$ shows much lower variability on the wide range from 40 $m$ ($M = 1$) to over 1 $km$ ($M > 25$). Besides, the reduction of STDEV of $F_s$ from 40 $m$ to 120 $m$ is very small, and the overall reduction rate of $F_s$ is also lower compared with $F_r$ (slopes at -0.21, -0.17 and -0.18 respectively). This indicates that compared with $F_r$ , small-scale, local variability of $F_s$ is
relatively low. However, $F_s$ is controlled by both sea ice thickness and snow distribution, and it shows comparable variability as $F_r$ at larger scales. Compared with $F_s$ and $F_r$ , $h_s$ shows the lowest overall variability. The decrease of STDEV of $h_s$ with scaling is also the fastest, with slopes at: -0.43, -0.40 and -0.40. This indicates that at local areas, the snow depth is relatively more homogeneous at larger scales with respect to freeboards.





## 3.2 Basin-scale analysis of radar freeboard scaling

We extend the analysis of consistency between OIB and CS-2 to available OIB data on the basin-scale. Similar to regions in Fig. 3, we carry out analysis for all local regions (each of $37.5 \times 37.5 \ km^2$) with good OIB coverage and collocating CS-2 measurements, and compute the scaling of $F_r$ for each local region (similar to Sec. 3.1).

5 Sample variances (VAR) of both CS-2 and OIB for collocating local regions are shown in Fig. 4.a (original sample variances with $M = 1$ for OIB) and 4.b (400 $m$ with $M = 10$ for OIB). Each point represents a local region, and is colored according to the MYI fraction of the local region. Specifically, data from Korosov et al. (2018) are adopted for 2013 to 2017, and Ye et al. (2016) for 2011 and 2012 when the former is not available.

We compute the linear fitting between the VAR of OIB and CS-2 for MYI (MYI fraction > 90%, red line) and FYI (MYI 10 fraction < 10%, blue line) dominated regions. As is shown, there exists statistically significant correlation between OIB and CS-2 ($p < 0.01$) for both FYI and MYI at OIB's original resolution ($M = 1$), as well as 400 $m$ ($M = 10$).

We also carried out similar analysis with AWI's per-track CS-2 product. In terms of the CS-2 freeboard retrieval, there is more strict waveform filtering in AWI's protocol as compared with ESA. Accordingly, the radar freeboards from AWI show lower variability as compared with ESA. When using $M = 10$ for OIB, we also witness much larger variance of $F_r$ in AWI's 15 data than OIB. However, there also exists statistically significant fitting ($p < 0.01$ for both FYI and MYI) between VAR of $F_r$ from AWI's product and that of OIB (not shown), which is consistent with the analysis of ESA CS-2 product.

This result with basin-scale observations confirms that CS-2 generally shows larger variability of $F_r$ than OIB. For $M = 1$, the intercept of the linear fitting of VAR both FYI and MYI are 0.019 and 0.04 $m^2$ respectively (Fig. 4.a). For $M = 10$, they are 0.02 and 0.045 $m^2$ (Fig. 4.b). By assuming the additive nature of the CS-2 noise, we deduce that the noise levels of ESA 20 CS-2 $F_r$ product for FYI and MYI are about 14 $cm$ and 20 $cm$. The noise levels of AWI's product are about 10 $cm$ and 14 $cm$, respectively. This estimation is slightly higher than existing studies of less than 10 $cm$ as in Ricker et al. (2014).

In Fig. 4.c we show the analysis with data from collocating tracks between CS-2 and airborne campaigns of OIB and CryoVEx. Since no direct measurement of $F_r$ is available from CryoVEx, we follow the density settings in altimetry (Sec. 2) and approximate $F_r$ with 1/10 of the total thickness. In total, 11 OIB tracks and 7 CryoVEx tracks are included in the analysis. 25 For CS-2, only samples on these collocating tracks are used for analysis. Similar to previous analysis, we divide the tracks into local segments, and show the along-track 400 $m$ average for OIB/CryoVEx to align with CS-2's along-track footprint size.

The across-track footprint size for $F_r$ is different by over 40 times between OIB (or CryoVEx) and CS-2. Compared with analysis based on monthly collocating CS-2 data (Fig. 4.b), there is a larger spread between the variance by collocating tracks of OIB/CryoVEx and CS-2, mainly due to reduced CS-2 sample count and limited representativeness. However, there still 30 exists statistically significant ($p < 0.01$) correlation between VAR of CS-2 and that of OIB/CryoVEx.

The analysis indicates that although there is a relatively high noise level of CS-2 freeboard products, the overall variability is consistent with high-resolution, airborne measurement from OIB and CryoVEx. For a given location, if the sea ice cover with larger (smaller) variability of $F_r$ on the small spatial scale, CS-2 also consistently produces $F_r$ samples that indicate higher (lower) variability. This provides us an indirect method for estimating the variability of $F_r$ at high resolution (e.g., 40 $m$ for



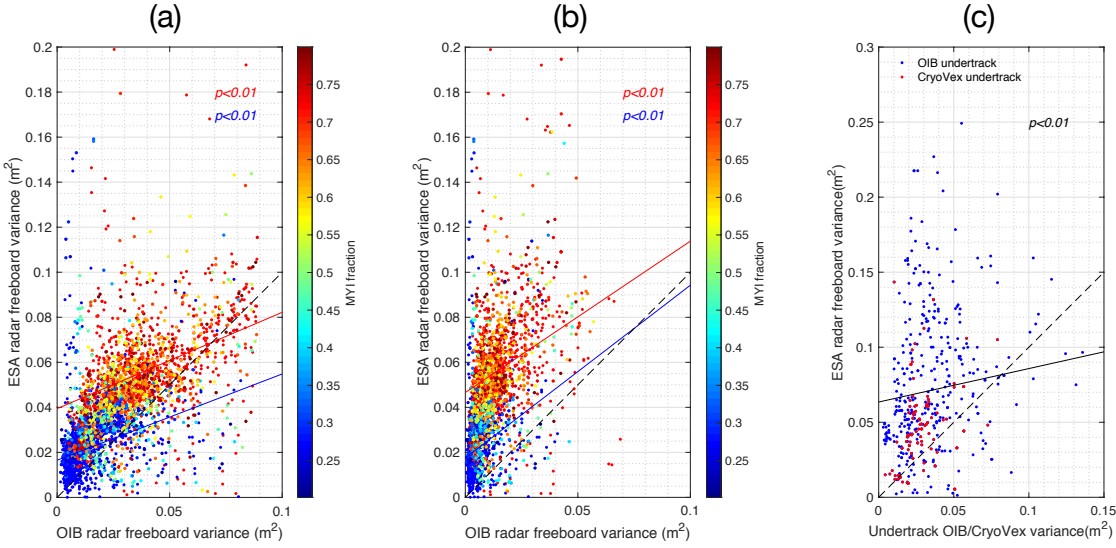

**Figure 4.** Sample variance of $F_r$ between OIB, CryoVex and CS-2. Panel a and b use the sample variance as measured by original OIB resolution ($40\ m$) and that by OIB on $400\ m$ resolution (local averaging with $M = 10$). Panel c shows the comparison of the collocating tracks of OIB and CVex with CS-2 on the scale of CS-2's along-track footprint of $400\ m$.

OIB) using CS-2 samples which are relatively lower in resolution. By using ESA CS-2 $F_r$ product and following Fig. 4.a, we deduce the variability at OIB scale of $40\ m$ ($STDEV_{40m}$) using CS-2 samples ($STDEV_{cs2}$) as follows:

$$STDEV_{40m} = 0.45 \cdot STDEV_{cs2} + 0.10 \quad, for\ MYI \tag{5}$$

$$STDEV_{40m} = 0.24 \cdot STDEV_{cs2} + 0.07 \quad, for\ FYI \tag{6}$$

5    It is worth to note that the actual distribution of $F_r$ at OIB scale is not directly reproducible with CS-2 $F_r$ variability. However, the 2nd-order moment of the distribution (STDEV or VAR) of $F_r$ on OIB scale can be indirectly estimated.

### 3.3  Scaling analysis of snow freeboard

In this section we compare the statistical scaling of $F_s$ measurements by OIB and ICESat. Since there is no temporally col­locating data, we compare the statistical consistency of $F_s$ variability and scaling. Similar to $F_r$ , the analysis of $F_s$ scaling

10  is also based on local regions. For comparison, ICESat attains bi-monthly basin coverage in autumn and winter, while most OIB campaigns are carried out during high winters in the western Arctic. Therefore we focus their measurements during Feb., Mar. and Apr. in the western Arctic, and differentiate between FYI and MYI. Each local region for computing the STDEV and scaling consists of about 7'073 to 24'100 ICESat bi-monthly samples for each of the 5 winters between 2003 and 2008. Since





there is no MYI fraction data product available for these years, we adopt the ice type as specified by ICESat and OIB datasets. In total, 70'673 and 96'528 local regions for FYI and MYI are recorded by ICESat. Similar to ICESat data, all OIB tracks within the same year for a local region is treated as a unit for computing STDEV.

Fig. 5 shows the STDEV of $F_s$ and its scaling. Based on original sample resolution for both ICESat (65 $m$ in diameter) and OIB (40 $m$ in diameter), the modes of STDEV distribution for ICESat and OIB are 0.175/0.105 $m$ (MYI/FYI) and 0.165/0.093 $m$ (MYI/FYI), respectively (blue lines in Fig. 5.a and c). Both OIB and ICESat show larger variability of $F_s$ for MYI than FYI (about 50% higher STDEV). The effective footprint for ICESat is about 3'320 $m^2$, and the area covered by OIB is lower at 1'260 $m^2$ (see Fig. 1.b). On the original footprint sizes, ICESat shows slightly higher variability. This results aligns well with the quantitative variability and uncertainty of these two products. First, after eliminating the effect of random error of ICESat ($\sigma = 5cm$) from its sample variance, the mode of STDEV distribution for ICESat is reduced from 0.175 $m$ and 0.105 $m$ to 0.168 $m$ and 0.092 $m$. These values are close to those observed by OIB. As previously shown in the scaling analysis, there is only slight decrease of $F_s$ variability from 40 $m$ to 80 $m$ (Fig. 3.c). Therefore, on the native footprints, we consider that the variability as measured by ICESat and OIB are consistent. ICESat measurements precede those of OIB by over 5 years, and we do not witness significant change in $F_s$ variability across the Arctic basin during this period.

In Fig. 5.a and c, we also show the PDF of STDEV under local averaging. As is shown, at coarser spatial scales, the overall variability of $F_s$ drops for both OIB and ICESat. In order to accommodate the differences in footprint sizes and spatial coverage by OIB and ICESat, we study two stages for scaling analysis: (1) $M = 20$ for OIB for comparison with $M = 10$ for ICESat, and (2) $M = 60$ for OIB for comparison with $M = 30$ for ICESat. For the first stage, OIB coverage is 800 $m$ of 20 consecutive and non-overlapping "dots" of 40 $m$ in diameter, while ICESat footprint is 10 consecutive "dots" of 1'600 $m$ apart, with each dot covering about 3'300 $m^2$. The aggregated footprint size for OIB is comparable to ICESat (25'000 $m^2$ for OIB and 33'000 $m^2$ for ICESat). For the first stage, the modes of STDEV for OIB and ICESat are both at 0.1 $m$ and 0.06 $m$ for MYI and FYI respectively. For the second stage, the modes are 0.06 to 0.07 $m$ (MYI) and 0.04 to 0.05 $m$ (FYI). Here we ignore the random error's effects, since the influence on sample STDEV is less than 2 $mm$ for $M > 20$.

We also compare the scaling behavior of $F_s$ in Fig. 5.b (for OIB) and d (for ICESat). For each local region, the slope is computed based on 40 $m$ to 1200 $m$ ($M = 1$ to $M = 30$) for OIB, and 65 $m$ to 1750 $m$ ($M = 1$ to $M = 10$) for ICESat. For OIB, the modes of slope PDF for MYI and FYI are -0.2 and -0.16. For ICESat, they are of similar values at -0.2 and -0.16. Also the distribution of slopes are quite similar with similar standard deviation of 0.07.

To summarize, the general behavior of $F_s$ scaling by ICESat and OIB are consistent, with similar value in variability and scaling behaviors. Although ICESat observations are several years before OIB campaigns, we do not observe significant changes in the variability of $F_s$ and its scaling. It is also noted that both footprint size and spatial coverage are important to the comparison of scaling behaviors. Both smaller footprints and wider coverage (through more heterogeneity) could induce larger variability in $F_s$ .

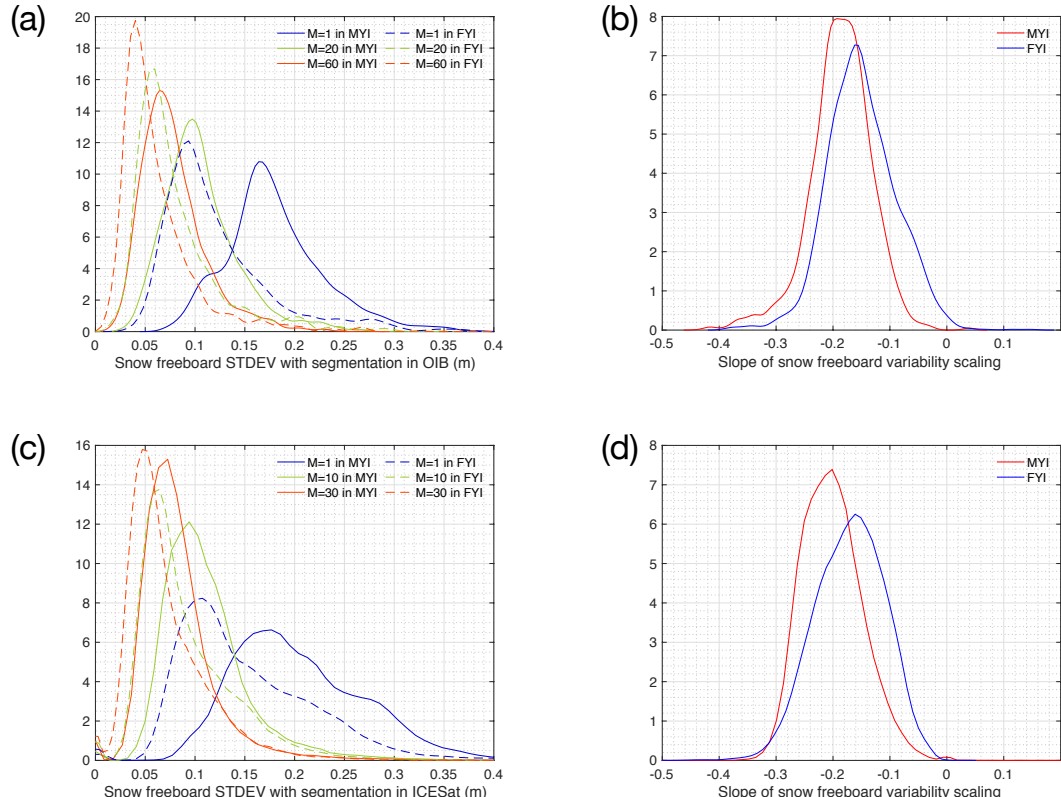

**Figure 5.** Variability of $F_s$ and its scaling for OIB (a and b) and ICESat (c and d) in the western Arctic.

### 3.4 Covariability of snow depth and freeboard

In this section we study the covariability between $h_s$ and freeboard measurements (both $F_s$ and $F_i$). We mainly use OIB datasets for this analysis, since both $h_s$ and $F_s$ are available, and they are measured and retrieved independently. The measurement errors ($e$ and $\epsilon$) are formulated as below, with differentiation between the observed (denoted by $|_o$) and inherent physical states (denoted by $|_p$) of the parameters.

$$F_s|_o = F_s|_p + e_{ATM} + \epsilon_{ATM} \tag{7}$$

$$h_s|_o = h_s|_p + e_{SR} + \epsilon_{SR} \tag{8}$$

$$F_i|_o = F_s|_o - h_s|_o \tag{9}$$

$$= F_i|_p + (e_{ATM} - e_{SR}) + (\epsilon_{ATM} - \epsilon_{SR}) \tag{10}$$





For covariability between $F_s$ and $h_s$, we show in Eq. 11 the deduction of covariance between $F_s|_p$ and $h_s|_p$. Hereby we assume: (1) statistical independence of errors ($e_{ATM}$, $e_{SR}$, $\epsilon_{SR}$ and $\epsilon_{ATM}$) from physical parameters ($F_s|_p$ and $h_s|_p$), (2) statistical independence relationships between errors, and (3) no local variability of bias terms. Then we deduce that the sample covariance estimated from observations between $F_s|_o$ and $h_s|_o$ is an un-biased estimator for the true, physical covariance. Fig.
6 shows the PDF of sample covariance between $F_s$ and $h_s$ for local regions with good coverage (panel a and b for 40 $m$ and 800 $m$, respectively). At 40 $m$, 95 % of all local regions show statistically significant positive covariance (hence correlation), while at 800 $m$, still 90 % retain positive covariance. This dominant feature is consistent with the physical perspective (Eqs. 2) that the thicker snow cover induce higher total freeboard.

$$Cov(F_s|_o, h_s|_o) = Cov(F_s|_p + e_{ATM} + \epsilon_{ATM}, h_s|_p + e_{SR} + \epsilon_{SR})$$

$$= Cov(F_s|_p, h_s|_p) \tag{11}$$

For $F_i$ and $h_s$, we also deduce the covariability as in Eq. 12. Since $F_i$ is a derived parameter from $F_s$ and $h_s$ in OIB, the estimation of covariance between $F_i$ and $h_s$ may be biased by random errors. Specifically, random error in $h_s$ measurements casts a positive offset on $Cov(F_i|_o, h_s|_o)$, by $\frac{2.5 \times 10^{-3}}{M} m^2$ where $M$ is the sample count for along-track averaging. This implies that under the alternative hypothesis of negative covariance between $h_s$ and $F_i$, using sample covariance without this correction
would increase the chance of Type-I error for the testing.

$$Cov(F_i|_o, h_s|_o) = Cov(F_i|_p + (e_{ATM} - e_{SR}) + (\epsilon_{ATM} - \epsilon_{SR}), h_s|_p + \epsilon_{SR})$$

$$= Cov(F_i|_p, h_s|_p) - \sigma_{SR}^2 \tag{12}$$

Fig. 7.a and b show the distribution of derived $Cov(F_i|_p, h_s|_p)$ at 40 $m$ and 800 $m$, respectively. At 40 $m$ scale, for MYI-dominated, FYI-dominated, and mixed ice type regions, we have 97 %, 72 % and 91 % regions with negative covariance
(statistically significant at 95% confidence level). However, at 800 $m$, only 30 %, 8 % and 12 % regions shows negative covariance, respectively. This result indicates that at small scale, there is complementary relationship between snow depth and ice freeboard. This is probably due to sea ice and snow interaction at small spatial scale. First, sea ice topographical features are more prominent at smaller spatial scales, which affect snow accumulation and result in deeper snow for local places with lower ice freeboard and thinner ice. On MYI with thicker ice and rougher topography, the negative covariability is more pronounced
than FYI. Second, since OIB campaigns are carried out during the high winters in the Arctic, thicker snow may have induced lower ice thickness growth during the whole winter, which is a potential contributor to the thinner ice and lower freeboard.

On the other hand, the disappearance of this negative correlation at larger spatial scales indicates that this phenomena is mostly dominant on small scales. At larger scale, snow-ice interaction due to sea ice topography is less dominant, with much fewer regions showing negative covariance. Besides, at 800 $m$, certain regions (about 29 % for FYI and 21 % for mixture)
show significant positive covariance between $F_i$ and $h_s$. Difference in snowfall accumulation may be the reason: compared





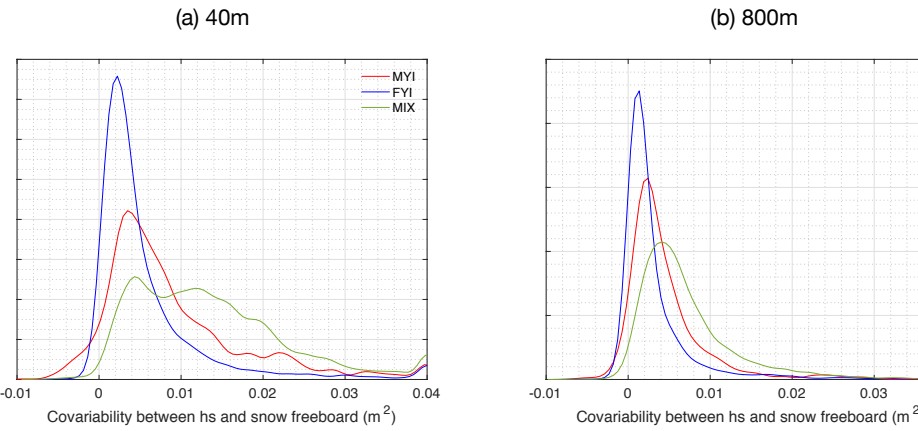

**Figure 6.** Covariability between $F_s$ and $h_s$ on 40 $m$ (a) and 800 $m$ scale.

with younger ice, relatively older ice (with larger $F_i$) are exposed to longer and heavier snowfall accumulation during autumn, resulting in thicker snow and large $h_s$ [see Fig. 4 of Boisvert et al. (2018)].

Furthermore, we compute for each local region the spatial scale on which the negative covariance becomes statistically insignificant. For a local region, we define the critical scale ($s$) of snow distribution as the scale: (1) on which there is statistically significant negative covariability between $h_s$ and $F_i$, and (2) beyond which the negative covariability is not evident. For all the local regions with sufficient OIB coverage ($N$ over 50-th percentile), the distributions of $s$ for MYI, FYI and mixture region are shown in Fig. 7.c. For FYI, about 28 % shows no negative covariance at 40 $m$, with the 90-th percentile of $s$ at 280 $m$. For MYI dominated regions and regions with MYI-FYI mixture, there exists a well-defined mode for $s$ at 80 $m$, and a long-tail distribution (90-th percentile at 1120 $m$ and 520 $m$, respectively). In Fig. 7.d we also show the spatial distribution of $s$ in the Arctic basin. As is shown, there exists large spatial variability of $s$ both locally and across the basin, and regions with MYI usually feature much larger $s$. Regions where the roughest MYI manifests (north of CAA and Greenland, and certain regions in Beaufort Sea with remnant MYI) show the largest $s$. Compared with FYI ($s$ at about 40 $m$), thick MYI show much larger $s$ (over 500 $m$ for certain regions), which is evident of snow cover's complementary effect on reducing the sea ice cover's overall variability.

## 4 Discussion and Conclusion

In this article we examine the variability, its scaling and consistency among various remote sensing methods for Arctic sea ice, including airborne (OIB), radar altimetry (CS-2) and laser altimetry (ICESat). Analysis with collocating measurements by OIB and CS-2 shows: although CS-2 products contain a higher noise level on the per-sample scale, there is statistical consistency between variability of $F_r$ as measured by OIB and CS-2. The noise level of ESA's CS-2 baseline-C $F_r$ product is estimated at





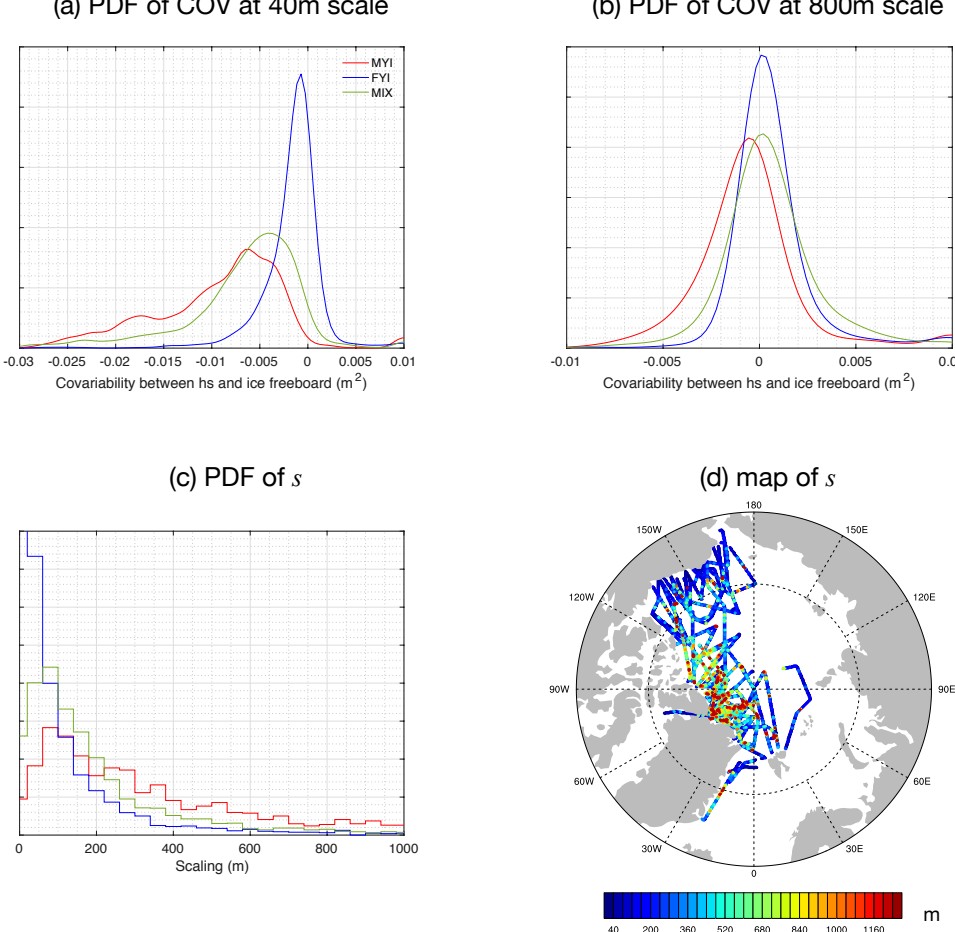

**Figure 7.** Covariability between $F_i$ and $h_s$ at 40 $m$ (a) and 800 $m$ scale (b). Distribution of sample-estimated covariance between $h_s$ and $F_i$ are shown (with correction for $\epsilon_{SR}$). Critical spatial scale ($s$) for negative covariability between $h_s$ and $F_i$ is shown in panel c (PDF) and d (geolocations).





14 to 20 $cm$, which is larger than current estimations. On the other hand, there is general consistency for both $F_s$ variance and its scaling by OIB and ICESat. We do not observe significant changes of these statistics from ICESat (2004 ~ 2008) to OIB (2009 on). Furthermore, by using OIB data, we also discovered wide-spread negative covariability between ice freeboard and snow depth at small spatial scales. This covariability generally diminishes at larger scales, indicating that the dominant role of

snow-ice interaction on local scales. The critical scale for the covariability is estimated using basin-wide OIB measurements, showing shorter ranges for FYI and much longer ranges for MYI with a long-tail distribution. The largest scale (over 500 $m$) is generally witnessed on the roughest MYI in the Arctic basins.

**Evaluation of variability**

For the comparison between OIB and CS-2, existing works mainly focus on the consistency of the mean freeboard or ice

thickness. As reported by Kurtz et al. (2014); Xia and Xie (2018), there is low correlation of $F_r$ between OIB and CS-2, using either gridded data or collocating tracks. Both retracking and geophysical corrections are shown to play an important role to reduce the difference (or bias) between OIB and CS-2 (Xia and Xie, 2018; Yi et al., 2018). Usually the limited representativeness is attributed as the main cause of low correlation, which arises from differences in footprint size and spatial coverage. In this work, instead of mean values, we analyze second-order statistics (i.e., the variability) and its scaling for radar freeboards.

Although CS-2 shows much higher $F_r$ variance than OIB even at 400-$m$ scale, the statistically significant relationship is witnessed. This result implies that the small-scale variability of $F_r$ can be quantitatively informed with CS-2 samples, in spite of high noise level of CS-2.

**Specifics about OIB**

The analysis of OIB, CS-2 and ICESat data indicates that all the factors contribute to the variability and its scaling, including

measurement footprints, noise levels, and spatial coverage. Except for the analysis of collocating tracks, we treat (daily) OIB campaigns and monthly CS-2 tracks as collocating sources of measurement. Since sea ice drift is prominent on the monthly scale, we also carried out analysis with CS-2 sample points under drift corrections (e.g., using NSIDC drift). We didn't notice evident change in the results for the analysis of variability, its scaling, and the estimation of CS-2 noise levels.

As for the data independency in OIB, since $F_s$ and $h_s$ are directly retrieved with ATM and SnowRadar respectively, they are

considered as independent data. However, $F_i$ , $F_r$ and in turn $h_i$ , are computed indirectly from $F_s$ and $h_s$ . The random error in $F_s$ and $h_s$ would cause uncertainty in both the variability and the covariability of these derived parameters. The analysis of variability scaling is less affected by random errors, since with a larger sample count the noise level decreases fast. The covariability between $F_i$ and $h_s$ , however, can be shifted by noise in $h_s$ measurements, and the effects are accounted for computing covariances in Sec. 3.4.

Existing works, including Kwok et al. (2017), carried out evaluation of various OIB datasets. For example, Kwok et al. (2017) shows general agreement, but systematic differences among $h_s$ retrievals, especially when the snow cover is thick (Fig. 4 and 5 of the reference). Since the validation showed good correlation of all $h_s$ products in Kwok et al. (2017), we consider that adopting alternative OIB datasets will not qualitatively alter the results in this study. Specifically, the products used in our study (IDCSI4 and IDCSI2) have underestimation of $h_s$ , so we expect that a thicker $h_s$ product would result in higher





variability in $h_s$ , as well as more evident negative covariance between $F_i$ and $h_s$ . Comparative studies of various OIB products can be carried out in the future when they become available, following the methodology proposed in this study.

**Effect of covariability on altimetry**

The snow cover is a major source of uncertainty for both types of satellite altimetry. The variability of the measured free-
board, as well as the covariability between snow depth and freeboard can be exploited to improve satellite altimetry. For laser altimetry, the positive covariability between $F_s$ and $h_s$ is present across spatial scales (40 $m$ to over 1 $km$), which covers the typical footprint size of ICESat and ICESat-2. This covariability is also reported by other works, including Kwok et al. (2011) (4 $km$ scale) and Zhou et al. (2018) (scale ranging from 40 $m$ to 240 $m$). Furthermore, in Zhou et al. (2018), nonlinear fittings between $h_s$ and $F_s$ (based on 40 $m$ scale covariability) are utilized for the combined retrieval of $h_s$ and $h_i$ with L-band passive
radiometry with laser altimetry. For laser altimetry with prescribed snow depth estimations, this positive covariability can also be utilized: (1) to avoid the artificial reduction of snow depth [in favor of non-negative ice freeboard as in Kwok and Cunning-ham (2008)], and (2) for the retrieval of ice thickness distribution with altimetric samples. Specifically, the covariability at the satellite sensor's footprint scale (65 $m$ diameter for ICESat) should be utilized. On the other hand, for radar altimetry for which $F_i$ is the major target for retrieval, the negative covariance between $h_s$ and $F_i$ only manifests at small scales. Whether there is
prevalent negative covariability at the footprint size of CS-2 is subjected to further study.

Based on the statistical fitting between the variability of OIB and CS-2, we derive an indirect method in Sec. 3.2 for estimat-ing the "real" variability at small scale (e.g., 40 $m$) using CS-2 samples. Then, the mean value of freeboard (or thickness) from CS-2 can be combined with this derived variability, to better inform applications that utilize CS-2 datasets. These include sea ice thickness assimilation applications [Chen et al. (2017) and Blockley and Peterson (2018) among others], which potentially
utilize mean thickness, mean freeboard, and freeboard samples from CS-2. However, statistical fitting specific to the CS-2 product should be utilized, since noise levels may differ among various products (see Sec. 3.2). It is worth noting that: it is not generally possible to directly estimate the freeboard distribution on small-scale with CS-2, given: (1) the large footprint size of CS-2, and (2) its relatively high noise level at per-sample scale. For example, in King et al. (2018) the N-ICE2015 campaign which surveyed local regions with second-year ice floes mixed with first-year floes and leads, the multi-modal distribution
(from high-resolution mapping) is absent in collocating CS-2 measurements (Fig. 10 of the reference). The physical features of freeboard distribution on the CS-2 footprint size, such as multi-modality, depend on the specific sea ice type (age) mixture and mixture scale of the region. In this regard, other datasets, such as SAR images, ice type mixture maps (Korosov et al., 2018), lead (history) maps (Zhou et al., 2017) can be utilized for a more holistic view of the ice cover.

**Snow-ice interaction**

There exists negative covariability between $F_i$ and $h_s$ at small-scale, which is consistent with in-situ measurements. On small scale, snow cover tends to complement sea ice topography (Sturm et al., 2002a), and the main factor may be snow accumulation through its interaction with topographic features such as ridges and refrozen ponds. Besides, snow depth also feature variabilities due to snow's own processes (other than those governed by ice), such as interaction with wind. On the other hand, due to thermal insulation of snow cover, sea ice thickness growth may also be hindered by thicker snow, resulting
in "thick snow - thin ice" relationship. Compared with previous works which mainly are based on in-situ measurements, this



study, by utilizing OIB data, reveals the critical spatial scale for the covariability of snow distribution due to interaction with sea ice. The common spatial scale for the negative covaribility is below 40 $m$ for FYI, and around 80 $m$ with a long-tail distribution for MYI.

The understanding of the dynamical and thermodynamical mechanisms that govern the statistical behaviors would requires

efforts from both modeling and observational aspects. Current state of the art sea ice models, including those in climate studies, usually contain major thermodynamic and dynamic processes of the sea ice cover, but many still lack snow related ones such as snow (re)distribution and prognostic snow stratigraphy. Considering the complex and important roles of snow on modulating air-sea interaction (Sturm et al., 2002b; Abraham et al., 2015), snow distribution and interaction with ice topography should be accounted for by refining vertical resolution of the sea ice and snow cover, as well as better parameterizations for

unresolvable scales. The airborne and in-situ observations, including the statistics of variability and covariability in this study, can be utilized to validate models and parameterization schemes. On the other hand, systematic observation during the freeze-up season is needed, in the pursuit of quantitative attribution to the statistics in snow distribution. Multi-scale and process-oriented observational campaigns such as MOSAiC (Alfred-Wegener-Institut) could potentially shed more light on the snow cover's key processes and its complex interaction with sea ice.

*Acknowledgements.*   The authors would like to thank the editors and referees for their invaluable efforts in improving the manuscript. This work is partially supported by National Key R&D Program of China under the grant number of 2017YFA0603902 and the General Program of National Science Foundation of China under the grant number of 41575076. The authors would also like to thank the data providers for the open access of CryoSat-2, ICESat, Operation IceBridge and CryoVEx datasets.

*Data availability.*   OIB datasets are provided by NASA National Snow and Ice Data Center Distributed Active Archive Center, Boulder,

Colorado USA. OIB IDCSI4 dataset is accessible at: https://doi.org/10.5067/G519SHCKWQV6 (last access: 14 September, 2019). OIB ID-CSI2 quick-look dataset is accessible via: https://daacdata.apps.nsidc.org/pub/DATASETS/ICEBRIDGE/ (last access: 14 September, 2019). ESA CS-2 along-track freeboard (L2i, baseline C) product is available at: https://science-pds.cryosat.esa.int/ (last access: 13 September, 2019). AWI CS-2 along-track freeboard is present in AWI CS-2 Sea Ice Thickness (version 2.1) daily trajectory L2P product, accessible via: ftp://ftp.awi.de/sea_ice/product/cryosat2/v2p1/nh/ (last access: 14 September, 2019). CryoVEx is from ESA Earth Observation Campaigns,

accessed at: ftp://calval-pds.cryosat.esa.int/ (last access: 14 September, 2019). ICESat data is provided by NASA National Snow and Ice Data Center Distributed Active Archive Center, Boulder, Colorado USA, via: https://doi.org/10.5067/SXJVJ3A2XIZT (last access: 14 September, 2019).

*Competing interests.*   The authors declare no conflict of interest



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
