# Peer review of "Variability Scaling and Consistency of Airborne and Satellite Altimetry Measurements of Arctic Sea Ice"

_The Cryosphere, 2019_

## Referee Comment (RC1) · Anonymous Referee #1 · 8 Nov 2019

General comments

The data used for the study are well presented. But the diagnosis performed are sometimes not enough explained or do not cleary serve the objective. Moreover, it is not fully clear to me which scientific objective is pursued with this analysis. Perhaps two many numbers dilute the aim...

Specific comments

[page#3, row#25-27] "On the other hand, by adopting the same the geophysical corrections of CS-2, Yi et al. (2018) effectively aligns the retrieved freeboard across CS-2 products and greatly reduces the systematic differences." Could you clarify ?

[Figure]

[page#4, row#22] "sea-surface height correction" "correction" is not relevant here, I would simply write "sea surface height"

[page#4, row#24] "in the freeboard estimation of the sea floes" I would remove "of the sea floes"

[page#4, row#25] "the freeboard uncertainty that is associated with SSH correction" SSH is not a correction. I would say SSH estimation or SSH retrieval

[page#6, row#33-34] "Since SSH height information are shared among freeboard data, we treat this uncertainty as bias and ignore it in the scaling analysis" I don't fully agree. Sea level anomaly (SLA) interpolation between the leads includes mean sea surface (MSS) error which is not necessary a bias. At the scale you are focus on, the explanation you give on page#8 row#5-7 seems more appropriate.

[page#7, row#9&18&32] "SSH correction" Not appropriate. Do you mean geophysical corrections (troposphere, ionosphere, tide) ? Or Sea level estimation ?

[page#7, row#15] equation (4) I prefer the (equivalent) formulation by Kurtz 2014 $F\_i=F\_r+h\_s (1-c\_s/c)$

[page#11, row#14] "Since with random samples, the effects of [. . .] inhomogeneity are very limited" Could you explain ?

[page#12, fig#3] Could you explain how each point of the curve has been computed ? Does the size of the considered area change for each point ? (fig3a) On fig.3b do you change M for each point, leading to a scale = M*resolution ? I don't understand why STDEV is larger when averaging (3b vs 3a).

[page#12, row#5] "SSH correction" Not appropriate. Do you mean geophysical corrections (troposphere, ionosphere, tide) ? Or Sea level estimation ?

[page#12, row#11-12] It is not so easy to compare fig#3b and #3c as the y-scale are different.

[page#12, row#15-16] "However, Fs is controlled by both sea ice thickness and snow distribution, and it shows comparable variability as Fr at larger scales." But Fr(OIB) is linked to hs as it is estimated using Fs and hs.

[page#13, row#6] "Each point represents a local region" Does it mean that 1 point corresponds to the STDEV over 37.5x37.5km$^2$ ?

[page#13, row#10] "As is shown, there exists statistically significant correlation between OIB and CS-2" It seems not so significant to me... and it is even worse at 400m.

[page#13, row#11&15&30] p < 0.01 Could you explain what is p ? What does it mean ?

[page#13, row#13] "more strict waveform filtering in AWI's protocol as compared with ESA" Could you explain ? Is it linked to waveform classification or editing ?

[page#13, row#29-30] "However, there still exists statistically significant (p < 0.01) correlation between VAR of CS-2 and that of OIB/CryoVEx." I am not convinced that the correlation is significant...

[page#13, row#32-34] "For a given location, if the sea ice cover with larger (smaller) variability of Fr on the small spatial scale, CS-2 also consistently produces Fr samples that indicate higher (lower) variability." I don't understand this sentence ; could you clarify ?

[page#14, fig#4] What does mean p<0.01 ? On fig#3b it seems that OIB variance is almost killed when Fr is averaged over 10 points. It seems not inline with fig#3b.

[page#14, row#1-2] "By using ESA CS-2 Fr product and following Fig. 4.a, we deduce the variability at OIB scale of 40 m (STDEV40m) using CS-2 samples (STDEVcs2)" Could you explain the scientific interest to do so ?

[page#15, row#9-10] "First, after eliminating the effect of random error of ICESat ($\sigma$ = 5cm) from its sample variance" Could you explain how do you proceed ?

[page#15, row#31-32] "Both smaller footprints and wider coverage (through more heterogeneity) could induce larger variability in Fs" It not so clear to me how you can conclude this...

[page#16, row#1] "covariability" (also on pages#16-18) Do you mean covariance ?

[page#17, row#8] "that the thicker snow cover induce higher total freeboard" But over MYI, there could be ridges impacting Fs... I would have expect more correlation between low hs and low Fs over FYI.

[page#17, row#15] "Type-I error" Could you explain what does it mean ?

[page#17, row#21-22] "This result indicates that at small scale, there is complementary relationship between snow depth and ice freeboard." Could you explain what you mean ? I would say they are uncorrelated (opposite variations).

[page#18, row#13] "complementary effect" Could you explain what do you mean ? masking effect ?

[page#18, fig#6] Even if the covariance is positive, it is still around zero ; so I don't see any clear correlation between hs and Fs on this figure.

[page#21, row#7] "This covariability is also reported by other works, including Kwok et al. (2011)" I think that the graphs used in this paper from Kwok are more relevant and easier to understand.

[page#21, row#31] "snow cover tends to complement sea ice topography" Do you mean mask ?

Technical corrections

[page#3, row#25] "On the other hand, by adopting the same the geophysical corrections" There is probabbly one "the" to be removed.

---

## Referee Comment (RC2) · Anonymous Referee #2 · 12 Dec 2019

The paper presents a new and interesting view on covariability in radar and laser altimeter data of sea-ice and its snow cover. The paper is well written and results are presented in a clear way, so most of my comments below refer to what may be typos.

Detailed comments: P1L10: despite over 5 years' the time difference -> despite the over 5 year time difference

P2L12: perspective -> perspectives

P2L17: in-situ observations of sea ice concentration is I believe equally chellenging

P2L24: added?? I suppose you mean that the local SSH is subtracted from the local

floe's height

P2L28: too many "main/mainly"

The main backscattering plane mainly resides close to the surface of the snow cover, and the main target is the retrieval of the snowfreeboard (Fs). -> The main backscattering plane resides close to the surface of the snow cover, and the target is the retrieval of snow freeboard (Fs).

P2L34:corrected freeboard ice freeboard -> corrected freeboard is called ice freeboard

P3L1: effective freeboard -> apparent freeboard

P3L12+13: You should indicate which is for radar and which is for laser

P4L17: parameter and its -> parameters and their

P4L24: sea floes -> ice floes

Figure 1: In the top part there is significant discrepancy between variable names in figure and in the paper text. Check the use of capital letters, subscript and superscript. (h_subscript_s_superscript_*, C_subscript_s etc)

P6L6: utilizes -> performs

P6L26: footprint -> footprints

P6L29: You should refer to Eq 1 here.

P7L10: that radar -> that the radar

P7L15: This is where the discrepancy in nomenclature with Figure 1 is most apparent, for example c_subscript_s should be C_subscript_s according to figure 1). Also explain what c is (speed of light in vacuum?).

P7L18: all these three products adopt threshold -> att three products adopt a threshold

P7L22: under same -> under the same

P8L12: campaigns collocated -> campaigns have been colocated

P8L14-15: we use dataset -> we use a dataset

P8L20: under certain knowledge of -> under certain assumptions about

P8L26: several -> a few

P9L1: I suggest that you mention here that the ice drift will be discussed later. Also, there are many versions of the EASE grid. I gather that you are using a 12.5 kilometer EASE grid (or EASE2?)

P10L30: collocating -> colocation

P11L8: speed -> rate

P11L20: if slow -> if a slow

P11L20: can be induced -> can be inferred

P12L5: Speckle noise should reduce by sqrt(M) whereas SSH correction will have much longer autocorrelation length scale.

P12L8: in range -> in the range

P12L17-18: This indicates ...... Please explain better. Why does a faster decrease indicate that the snow is relatively more homogenous?

P13L7-8: Quite a mix of data sources, why? Have you checked for inconsistencies between the two datasets?

P13L20: Your estimates of noise levels would benefit from an estimate of the errorbars on the estimates. How accurate do you think your estimates are, and is 14 significantly larger that 10?

Figure 4: In the figure captions (or even better titles on the figures) you should be more clear about the difference between a) and b).

[Figure]

P14 and following you should change the 1000s separator "'" to "," or remove it. It is not necessary.

P15L8: This results -> This result

P16L4: You introduce two measurement error terms (e and epsilon). You should explain whet they are/represent.

P20L34: have underestimation -> may have underestimation

P22L4: requires -> require

P22L11. systematic observation -> systematic observations

---

## Referee Comment (RC3) · Anonymous Referee #3 · 16 Dec 2019

This is an interesting study which analyzes variability between CryoSat-2 and airborne data sets of freeboard from Operation IceBridge and CryoVEx. The analysis provides a useful look at scales of variability between the data sets which considers instrument noise, retrieval errors, and random error sources.

I found this to be a well-written manuscript which provides valuable information useful for comparisons between the satellite and airborne data sets, as well as interpretation of results from the data sets on an independent basis. I just have a few minor comments listed below:

P2 L30: I think it's a misconception to state that the radar penetrates a certain percent-

[Figure]

age of the way through the snow cover, rather what happens is that the return contains energy from the snow surface, snow volume, and ice surface and depending on the local conditions and tracker used this can lead to a bias in the retrieval of ice freeboard. P3 L27: I believe the bandwidth-limited range resolution of CS-2 also greatly contributes to the low correlation on a shot-to-shot basis. P14 L 7: There was actually temporally coincident data between OIB and ICESat in 2009, however the laser energy of ICESat had degraded substantially by this point to make data potentially problematic for a comparison such as this.

---

## Author Comment (AC1) · 11 Jan 2020

The authors would like to the referee for the invaluable comments. The following revisions and corresponding replies are made for each comment (in *green italic* font). Also, a revised version of the manuscript is provided as attachment. The replies to comments are as follows, and the revisions are highlighted in the manuscript in yellow.

**Reply to comments of Referee #1:**

*The data used for the study are well presented. But the diagnosis performed are sometimes not enough explained or do not cleary serve the objective. Moreover, it is not fully clear to me which scientific objective is pursued with this analysis. Perhaps two many numbers dilute the aim...*

Reply to the general comment: the authors thank the reviewer for pointing out the shortcomings of manuscript in conveying the main idea and results of the study. We would like to emphasize that the purpose of the study is to learn about the **variability** of the measured/retrieved sea ice thickness parameters by current airborne and satellite remote sensing. Thickness parameters include: **radar and laser freeboard**, and **snow depth**, and there are 3 topics for study: **variability, the scaling of variability, and co-variability** (between snow and freeboard). The analysis of variability may be greatly affected by measurement/data-processing errors, therefore, we pay attention especially how these uncertainties may affect the estimation of the true, inherent physical variability of these parameters. Therefore, the main purpose is two-fold: (1) on the observational technique side, we want to study the variability and scaling by different approaches; and (2) on the scientific side, the inherent physical behavior of variability scaling and implications of the processes. Inevitably, these two issues are entangled in many cases, including this study.

*Specific comments*

*[page#3, row#25-27] "On the other hand, by adopting the same the geophysical corrections of CS-2, Yi et al. (2018) effectively aligns the retrieved freeboard across CS-2 products and greatly reduces the systematic differences." Could you clarify?*
Reply: in order to clarify, we revise this sentence as: "*On the other hand, with the same geophysical correction of CS-2 and snow depth correction based on OIB SnowRadar, the mean freeboards from four CS-2 retrackers are all in agreement with ATM by 0.05 m (Yi et al. 2018)*".

*[page#4, row#22] "sea-surface height correction" "correction" is not relevant here, I would simply write "sea surface height"*
Reply: corrected.

*[page#4, row#24] "in the freeboard estimation of the sea floes" I would remove "of the sea floes"*
Reply: corrected by removing "*of the sea floes*".

*[page#4, row#25] "the freeboard uncertainty that is associated with SSH correction" SSH is not a correction. I would say SSH estimation or SSH retrieval*
Reply: corrected to "… associated with SSH estimations".

*[page#6, row#33-34] "Since SSH height information are shared among freeboard data, we treat this uncertainty as bias and ignore it in the scaling analysis" I don't fully agree. Sea*

*level anomaly (SLA) interpolation between the leads includes mean sea surface (MSS) error which is not necessary a bias. At the scale you are focus on, the explanation you give on page#8 row#5-7 seems more appropriate.*

Reply: the authors have revised this sentence to be more accurate, as follows: *Since along-track SSH's are usually constructed with observations on much larger spatial scales (over 100 km), their uncertainty is not considered in the variability scaling which involves local averaging within several kilometers.*

*[page#7, row#9&18&32] "SSH correction" Not appropriate. Do you mean geophysical corrections (troposphere, ionosphere, tide)? Or Sea level estimation?*

Reply: "*SSH correction*" is revised to "*mean SSH estimation and local sea-level correction*", which is a more precise statement.

*[page#7, row#15] equation (4) I prefer the (equivalent) formulation by Kurtz 2014 F_i=F_r+h_s (1-c_s/c)*

Reply: the equation (no. 4) is revised as indicated. The approximation to the coefficient for the correction (1-c_s/c) is taken as 0.25 according to Tilling et al. (2018, ASR). If it is computed according to Kurtz et al., (2014, TC) under the snow density assumption of $330 kg/m^3$, this coefficient is about 0.22. With either estimation, the major result of F_r (radar freeboard) is not affected.

*[page#11, row#14] "Since with random samples, the effects of [. . .] inhomogeneity are very limited" Could you explain?*

Reply: since there exists: (1) autocorrelation of nearby samples and (2) inhomogeneity of the sea ice cover (within 37.5km by 37.5 km) that is sampled, samples randomly chosen will be physically away from each other, which will attenuate the effect of BOTH local correlation AND inhomogeneity of the region where the samples are collected. With random sampling strategy, the variability is expected to decrease with respect to sample count for averaging (STDEV decreasing with the square root of M). This is indeed observed in Fig. 3a. Since it does not provide any further insight into the scaling of the parameters, this strategy is only provided as a baseline for reference.

*[page#12, fig#3] Could you explain how each point of the curve has been computed? Does the size of the considered area change for each point? (fig3a) On fig.3b do you change M for each point, leading to a scale = M*resolution? I don't understand why STDEV is larger when averaging (3b vs 3a).*

Reply: for the 3 subfigures in Fig. 3, each color corresponds a specific local region in Fig. 2(b-d), which are local regions that contain good OIB coverage. Each curve in Fig. 3 is produced with averaging several OIB samples, and the referee is correct that the scale is computed as linear to the sample count. For example, 800 m corresponds to 20 OIB samples, which involves averaging of 20 random (or local) samples for subfigure a (or b and c). The different behavior of variability decrease (slower decrease in b than a) is mainly due to two factors. First, the local positive correlation of the parameter (Hs or Fr or Fs) causes the variability to decrease slower. Second, the sea ice cover within the region of study is inherently inhomogeneous, so local averaging will not attenuate the variability that is present on the spatial scale that is larger than the sample footprint. The larger variability with local averaging (b) than random sampling (a) is actually fully expected. As mentioned in Sec. 3.1, when we use the randomized sampling, the STDEV decreases with the square root of sample count M, which follows the theory for independent samples quite well. On the contrary, local averaging is the typical manner of scaling analysis, which is more informative of the physical

variability of the sea ice cover. And indeed, it shows slower variability decrease with larger scale (subfigure b). This figure is further revised to align the range in y-axis (STDEV) for clearer viewing.

*[page#12, row#5] "SSH correction" Not appropriate. Do you mean geophysical corrections (troposphere, ionosphere, tide) ? Or Sea level estimation ?*
Reply: "SSH correction" is revised to "SSH estimation", which is the correct term as used in Ricker et al., (2014).

*[page#12, row#11-12] It is not so easy to compare fig#3b and #3c as the y-scale are different.*
Reply: the updated figure (also shown below) contains aligned scale for the y-axis.

[Figure]

*[page#12, row#15-16] "However, Fs is controlled by both sea ice thickness and snow distribution, and it shows comparable variability as Fr at larger scales." But Fr(OIB) is linked to hs as it is estimated using Fs and hs.*
Reply: the authors agree with the referee on the comment, therefore, the sentence is revised as follows: "*Fs, similar to Fr, is controlled by both sea ice thickness and snow depth, and it also shows comparable variability as Fr at larger scales.*"

*[page#13, row#6] "Each point represents a local region" Does it mean that 1 point corresponds to the STDEV over 37.5x37.5km²?*
Reply: the referee is correct that each point corresponds to variability (STDEV) over 37.5x37.5km².

*[page#13, row#10] "As is shown, there exists statistically significant correlation between OIB and CS-2" It seems not so significant to me. . . and it is even worse at 400m.*
Reply: the authors have revised the figures to include information of correlation and statistical fittings: $r$ for the correlation coefficient (specifically, Pearson's product moment correlation coefficient adopted here), the linear fitting relationship, and $p$-value for the statistical significance of the correlation. All the fitting lines contain significant correlation between the freeboard variances at 0.01 level (i.e., $p < 0.01$). See also below for further info.

*[page#13, row#11&15&30] p < 0.01 Could you explain what is p? What does it mean?*
Reply: $p$ is the probability value (*p-value*). In statistical hypothesis testing, it is frequently adopted for the testing the null hypothesis that there exists no relationship between the observed phenomena. If $p$-value is smaller than a certain significance level (of which 0.05 is adopted by many practices), then the corresponding correlation (indicated by $r$) is considered significant. A $p$-value lower than 0.01 (which holds for all the fittings in the figure) indicates that the positive correlation is highly significant.

*[page#13, row#13] "more strict waveform filtering in AWI's protocol as compared with ESA" Could you explain? Is it linked to waveform classification or editing?*

Reply: the relevant differences between ESA (Baseline-C) and AWI are formally listed as below: (1) ESA uses a 70% threshold for the amplitude of the first peak for floe echoes surface tracking point while AWI use a 50% threshold at the first maximum of radar-echo-power-based method to determine surface elevation for leads and floes; (2) in AWI's protocol, radar freeboards (on the per-waveform level) that are too large or small are not use for further processing, while in ESA's product, there is no such filtering (Bouffard et al., ASR, 2018). Therefore, this sentence is rewritten as: "*more strict waveform filtering in AWI's protocol than ESA's, in order to eliminate outliers according to radar freeboard values*".

*[page#13, row#29-30] "However, there still exists statistically significant (p < 0.01) correlation between VAR of CS-2 and that of OIB/CryoVEx." I am not convinced that the correlation is significant. . .*

Reply: the statistical information (*r* and corresponding *p*-value) is added in Fig. 3c. After re-checking, we do confirm that indeed the positive correlation is significant at 0.01 level (p=0.001). In order to further support our argument, we have included in Fig. 3 the analysis with AWI CS-2 Fr product, along-side ESA Fr (baseline-C). As shown in Fig. 3d/e/f (which compare against Fig. 3a/b/c), all the fittings show significant correlation at 0.01 level between AWI's product and OIB.

*[page#13, row#32-34] "For a given location, if the sea ice cover with larger (smaller) variability of Fr on the small spatial scale, CS-2 also consistently produces Fr samples that indicate higher (lower) variability." I don't understand this sentence; could you clarify?*

Reply: The sentence contained a grammatical error, and also for the sake of clarity, it is revised as: "*For a given location, if the sea ice cover shows larger (smaller) Fr variance on the small scale, CS-2 also consistently produces Fr samples that contain larger (smaller) variance*".

*[page#14, fig#4] What does mean p<0.01? On fig#3b it seems that OIB variance is almost killed when Fr is averaged over 10 points. It seems not inline with fig#3b.*

Reply: the explanation for *p*-value (which is an indicator for the significance level of the correlation) is added as reply above. Fig. 4 is also updated with related statistical information.

*[page#14, row#1-2] "By using ESA CS-2 Fr product and following Fig. 4.a, we deduce the variability at OIB scale of 40 m (STDEV40m) using CS-2 samples (STDEVcs2)" Could you explain the scientific interest to do so?*

Reply: the purpose of deducing Fr variability at small scale (i.e., OIB) is that this info is NOT generally available across the Arctic basin. This is mainly due to the spatially and temporally limited coverage of OIB. Given the established relationship with collocating OIB and CS-2 data, we can use this statistical relationship to attain a basin-scale Fr variability estimation with CS-2. This info can be further applied in many studies such as the thickness retrieval with CS-2 (Xu et al., Rem. Sens., 2018).

*[page#15, row#9-10] "First, after eliminating the effect of random error of ICESat (σ = 5cm) from its sample variance" Could you explain how do you proceed?*

Reply: in specific, the random error variance ($\sigma^2 = 25cm^2$) is subtracted from the sample variance, which is what we mean by "eliminating". The square root of the resulting value (as standard deviation) is then used to construct the PDF in Fig. 5c.

*[page#15, row#31-32] "Both smaller footprints and wider coverage (through more heterogeneity) could induce larger variability in Fs" It not so clear to me how you can conclude this. . .*

Reply: the authors apologize for the ambiguous statement. The revisions include: (1) a new paragraph (page 16, line 1 to 8) to simulate OIB with ICESat interval of 175m, accompanied by added sub-figures to Fig. 5 (on page 17); (2) the revised sentence to the following: "*Two factors affects the variability in Fs. First, with the increase in the aggregate footprint size, the variability decreases. Second, if the spatial coverage of samples increase while keeping the total footprint size constant, there is even more effective dampening in variability. This indicates that portion of variability on the local scale increases with wider coverage of local samplings.*"

*[page#16, row#1] "covariability" (also on pages#16-18) Do you mean covariance?*

Reply: the referee is correct that by "covariability" we mean the relation of co-varying between snow depth and freeboard, which is estimated through sample covariance.

*[page#17, row#8] "that the thicker snow cover induce higher total freeboard" But over MYI, there could be ridges impacting Fs. . . I would have expect more correlation between low hs and low Fs over FYI.*

Reply: the authors agree with the referee that over the small spatial scale, sea ice ridges greatly impact Fs. Furthermore, we would like to acknowledge two facts. First, snow distribution and ice freeboard (or topographic features) may be dominated by difference processes and hence feature independent variability. Besides the negative covariance between the two, both Fi and Hs have large part of variability that are not included in (or explained by) this negative covariability. Second, as pointed out by the referee, ice features such as ridges might pose extra problem to our analysis. For example, it is shown that on highly deformed ridges, OIB's snow radar may not be able to produce trustworthy retrieval of snow depth due to undetectable snow-ice and air-snow interfaces. In the manuscript we do note that specific versions of the OIB products (especially for snow depth) may quantitatively alter the results in this study. Therefore for revisions, we have added extra discussion in OIB part in Sec. 4, especially on this issue.

*[page#17, row#15] "Type-I error" Could you explain what does it mean?*

Reply: Type-I error is often referred to as *false positive*, which in this context corresponds to that the null hypothesis (that no negative covariability exists between Fi and Hs) can be more possibly falsely rejected, if the random error in Hs is not accounted for during the estimation of covariance.

*[page#17, row#21-22] "This result indicates that at small scale, there is complementary relationship between snow depth and ice freeboard." Could you explain what you mean? I would say they are uncorrelated (opposite variations).*

Reply: the authors would like to clarify that by "complementary" we actually mean the relationship of negative covariance between Fi and Hs: when Hs is higher (thicker snow), Fi is lower; when Hs is lower, Fi tends to be higher. This relationship is also reflected in various field studies. For example, in Sturm (2002) and Sturm et al. (2002), it is shown that small-scale interaction between the snow and ice produces thicker snow above pond ice (with lower Fi) than nearby hummocks (see Fig. 13 of the reference). As reported in Sec. 3.4 for 40m scale, the negative covariance between Fi and Hs is significant for 97% MYI regions and 72% FYI regions (0.05 significance level).

*[page#18, row#13] "complementary effect" Could you explain what do you mean? masking effect?*

Reply: the authors would like to clarify that by "complementary effect", we mean that the ice topography is attenuated by the snow's distribution, which masks out the overall ice topography and reducing the variability in freeboard.

*[page#18, fig#6] Even if the covariance is positive, it is still around zero; so I don't see any clear correlation between hs and Fs on this figure.*

Reply: the authors would like to point out that the PDF in Fig. 6 is based on sample covariance, which is in the unit of $m^2$. Therefore, the absolute value seemingly near 0 does not indicate that the correlation coefficient is low or the correlation is not significant. Actually, as indicated in Sec. 3.4, over 95% local regions show statistically significant positive correlation ($p<0.05$) at 40m scale, with over 90% at 800m. Since many works have shown specific examples of relationship between Fs and Hs (such as Kwok et al. 2011, Zhou et al., 2018), we do not show any example here.

*[page#21, row#7] "This covariability is also reported by other works, including Kwok et al. (2011)" I think that the graphs used in this paper from Kwok are more relevant and easier to understand.*

Reply: the authors agree with the referee that a specific example is more indicative of this positive correlation.

*[page#21, row#31] "snow cover tends to complement sea ice topography" Do you mean mask?*

Reply: the authors acknowledge that the referee is correct that by "complement" we mean that the snow cover tends to attenuate the sea ice topography, and snow distribution that masks out ice topographic features might be a dominant factor.

---

## Author Comment (AC2) · 11 Jan 2020

The authors would like to the referee for the invaluable comments. The following revisions and corresponding replies are made for each comment (in *green italic* font). Also, a revised version of the manuscript is provided as attachment. The replies to comments are as follows, and the revisions are highlighted in the manuscript in red.

**Reply to comments of Referee #2:**

*The paper presents a new and interesting view on covariability in radar and laser altimeter data of sea-ice and its snow cover. The paper is well written and results are presented in a clear way, so most of my comments below refer to what may be typos.*

*Detailed comments: P1L10: despite over 5 years' the time difference -> despite the over 5 year time difference*
Reply: corrected.

*P2L12: perspective -> perspectives*
Reply: corrected.

*P2L17: in-situ observations of sea ice concentration is I believe equally chellenging*
Reply: the authors agree with the referee's comment, and have revised it as: "… thickness parameters are challenging for observations …".

*P2L24: added?? I suppose you mean that the local SSH is subtracted from the local floe's height*
Reply: corrected by changing "added to the floe's height" to "subtracted from the floe's range".

*P2L28: too many "main/mainly"*
*The main backscattering plane mainly resides close to the surface of the snow cover, and the main target is the retrieval of the snowfreeboard (Fs). -> The main backscattering plane resides close to the surface of the snow cover, and the target is the retrieval of snow freeboard (Fs).*
Reply: revised by deleting unnecessary words of "mainly" and "main".

*P2L34:corrected freeboard ice freeboard -> corrected freeboard is called ice freeboard*
Reply: revised as indicated.

*P3L1: effective freeboard -> apparent freeboard*
Reply: corrected from "effective penetration" to "apparent penetration".

*P3L12+13: You should indicate which is for radar and which is for laser*
Reply: revised in the location referencing these two equations.

*P4L17: parameter and its -> parameters and their*
Reply: revised.

*P4L24: sea floes -> ice floes*
Reply: revised with deletion of "of sea floes".

*Figure 1: In the top part there is significant discrepancy between variable names in figure and in the paper text. Check the use of capital letters, subscript and superscript. (h_subscript_s_superscript_\*, C_subscript_s etc)*
Reply: this figure is fully revised to use the same variable names as the text.

*P6L6: utilizes -> performs*
Reply: corrected.

*P6L26: footprint -> footprints*
Reply: corrected.

*P6L29: You should refer to Eq 1 here.*
Reply: revised by adding the reference.

*P7L10: that radar -> that the radar*
Reply: corrected.

*P7L15: This is where the discrepancy in nomenclature with Figure 1 is most apparent, for example c_subscript_s should be C_subscript_s according to figure 1). Also explain what c is (speed of light in vacuum?).*
Reply: revised by adding necessary notation explanations in the text.

*P7L18: all these three products adopt threshold -> att three products adopt a threshold*
Reply: revised according to suggestion.

*P7L22: under same -> under the same*
Reply: corrected.

*P8L12: campaigns collocated -> campaigns have been colocated*
Reply: corrected.

*P8L14-15: we use dataset -> we use a dataset*
Reply: corrected.

*P8L20: under certain knowledge of -> under certain assumptions about*
Reply: corrected.

*P8L26: several -> a few*
Reply: corrected.

*P9L1: I suggest that you mention here that the ice drift will be discussed later. Also, there are many versions of the EASE grid. I gather that you are using a 12.5 kilometer EASE grid (or EASE2?)*
Reply : a sentence mentioning tests with ice drift correction is added by the end of the paragraph, as suggested by the referee. Besides, the referee is correct that we use EASE grid (instead of EASE2).

*P10L30: collocating -> colocation*
Reply: corrected.

*P11L8: speed -> rate*
Reply: corrected.

*P11L20: if slow -> if a slow*
Reply: corrected.

*P11L20: can be induced -> can be inferred*
Reply: corrected.

*P12L5: Speckle noise should reduce by sqrt(M) whereas SSH correction will have much longer autocorrelation length scale.*
Reply: the authors agree with the comments from the referee on the different rate of error decrease with scale.

*P12L8: in range -> in the range*
Reply: corrected.

*P12L17-18: This indicates ...... Please explain better. Why does a faster decrease indicate that the snow is relatively more homogenous?*
Reply : when a faster decrease is witnessed for a certain parameter (decrease speed closer to -0.5), then it indicates that there is lower heterogeneity of the parameter, since local average can more effectively attenuate its variability. The information above is added as the revision suggested by the referee.

*P13L7-8: Quite a mix of data sources, why? Have you checked for inconsistencies between the two datasets?*
Reply: the authors would like to clarify that the major reason of using both SICCI and U-Bremen MYI concentration products is that neither of them provides full coverage of 2011 to 2018. As reported by some studies, OSI-SAF MYI coverage product tends to underestimate the MYI extent, while U-Bremen product contains MYI concentration info, but tends to feature over-estimation (due to ambiguity with ASCAT on MYI coverage). Therefore, we use the combination of SICCI and U-Bremen product for the analysis. Since the two agrees quite well for regions where MYI or FYI dominates, we do not expect that the quantitative fittings of noise levels change much with the specific product we use.

*P13L20: Your estimates of noise levels would benefit from an estimate of the errorbars on the estimates. How accurate do you think your estimates are, and is 14 significantly larger that 10?*
Reply: the author would like to clarify that these estimations are based on statistical fittings of Fr variability of OIB and CS-2, and it is challenging to attain an estimation of the uncertainty. They are only provided as another estimations of the noise level of CS-2 for FYI and MYI, which are compared with other estimations (such as Ricker et al., 2014) in the paper.

*Figure 4: In the figure captions (or even better titles on the figures) you should be more clear about the difference between a) and b).*
Reply: Fig. 4 is full revised to include both ESA (a, b and c) and AWI (d, e, and f) CS-2 products. Also the caption is revised according to the referee's suggestion.

*P14 and following you should change the 1000s separator "'" to "," or remove it. It is not necessary.*

Reply: all the "'" are removed, as suggested.

*P15L8: This results -> This result*

Reply: corrected.

*P16L4: You introduce two measurement error terms (e and epsilon). You should explain whet they are/represent.*

Reply: the authors would like to point out that they have been introduced in Eqs. 3. A reference to Eqs. 3 is added here.

*P20L34: have underestimation -> may have underestimation*

Reply: revised as "may have underestimated".

*P22L4: requires -> require*

Reply: corrected.

*P22L11. systematic observation -> systematic observations*

Reply: corrected.

---

## Author Comment (AC3) · 11 Jan 2020

The authors would like to the referee for the invaluable comments. The following revisions and corresponding replies are made for each comment (in *green italic* font). Also, a revised version of the manuscript is provided as attachment. The replies to comments are as follows, and the revisions are highlighted in the manuscript in green.

**Reply to comments of Referee #3:**

*This is an interesting study which analyzes variability between CryoSat-2 and airborne data sets of freeboard from Operation IceBridge and CryoVEx. The analysis provides a useful look at scales of variability between the data sets which considers instrument noise, retrieval errors, and random error sources.*

*I found this to be a well-written manuscript which provides valuable information useful for comparisons between the satellite and airborne data sets, as well as interpretation of results from the data sets on an independent basis. I just have a few minor comments listed below:*

*P2 L30: I think it's a misconception to state that the radar penetrates a certain percentage of the way through the snow cover, rather what happens is that the return contains energy from the snow surface, snow volume, and ice surface and depending on the local conditions and tracker used this can lead to a bias in the retrieval of ice freeboard.*
Reply: the authors have corrected this sentence to be accurate, as follows: "… *the backscattering of radar signal may occur at the air-snow interface, through snow volume scattering, as well as at the snow-ice interface. With dry snow, it is usually assumed that radar signals can effectively penetrate the snow cover*".

*P3 L27: I believe the bandwidth-limited range resolution of CS-2 also greatly contributes to the low correlation on a shot-to-shot basis.*
Reply: the authors agree with the referee's comment. Indeed the range resolution of CS-2 is limited to about 0.23m due to limited bandwidth of 320MHz. This will be a contributing factor of low correlation on a shot-to-shot basis, and the sentence is revised for the sake of completeness. However, with local averaging, the effect of limited range resolution behaves like a random error and should diminish fast. But as investigated by many studies, even with local averaging, the correlation of mean Fr is still quite low with airborne collocating tracks. The authors acknowledge that it should be a factor, but possibly not a major one.

*P14 L 7: There was actually temporally coincident data between OIB and ICESat in 2009, however the laser energy of ICESat had degraded substantially by this point to make data potentially problematic for a comparison such as this.*
Reply: the authors have made revisions to make the sentence more accurate, as follows: "*Since there is no colocating data available between ICESat and OIB, …*"